# Potential Antioxidant and Anticancer Activities of Secondary Metabolites of *Nostoc linckia* Cultivated under Zn and Cu Stress Conditions

Khaled M. A. Ramadan [1,2,*], Hossam S. El-Beltagi [3,4,*], Sanaa M. M. Shanab [5], Eman A. El-fayoumy [5], Emad A. Shalaby [4,*] and Eslam S. A. Bendary [2]

1 Central Laboratories, Department of Chemistry, King Faisal University, P.O. Box 420, Al-Ahsa 31982, Saudi Arabia
2 Department of Biochemistry, Faculty of Agriculture, Ain Shams University, Cairo 11566, Egypt; esslam136@gmail.com
3 Agricultural Biotechnology Department, College of Agriculture and Food Sciences, King Faisal University, P.O. Box 420, Al-Ahsa 31982, Saudi Arabia
4 Biochemistry Department, Faculty of Agriculture, Cairo University, Gamma St, Giza 12613, Egypt
5 Department of Botany and Microbiology, Faculty of Science, Cairo University, Gamma St, Giza 12613, Egypt; sanaashanab@sci.cu.edu.eg (S.M.M.S.); emanelfayoumy@gmail.com (E.A.E.-f.)
* Correspondence: kramadan@kfu.edu.sa (K.M.A.R.); helbeltagi@kfu.edu.sa (H.S.E.-B.); dremad2009@yahoo.com (E.A.S.)

**Abstract:** The objective of the present study is to determine the antioxidant and anticancer activities of *Nostoc linckia* extracts cultivated under heavy metal stress conditions (0.44, 0.88, and 1.76 mg/L for zinc and 0.158, 0.316, 0.632 mg/L for copper). Phycobiliprotein, phenolic compounds, flavonoids, and tannins were measured. Active ingredients of extracts were evaluated by GC-mass spectroscopy. The obtained results revealed that higher zinc and copper concentrations showed growth inhibition while 0.22 mg/L (Zn) and 0.079 mg/L (Cu) enhanced growth, reaching its maximum on the 25th day. Increases in catalase, lipids peroxidation, and antioxidants, as well as tannins and flavonoids, have been induced by integration of 0.88 mg/L (Zn) and 0.316 mg/L (Cu). Elevation of Zn concentration induced augmentation of antioxidant activity of crude extract (DPPH or ABTS), with superior activity at 0.44 mg/L zinc concentration (81.22%). The anticancer activity of *Nostoc linckia* extract (0.44 mg/L Zn) tested against four cancer cell lines: A549, Hela, HCT 116, and MCF-7. The extract at 500 μg/mL appeared the lowest cell viability of tested cell lines. The promising extract (0.44 mg/L Zn) recorded the lowest cell viability of 25.57% in cervical cell line, 29.74% in breast cell line, 33.10% in lung cell line and 34.53% in the colon cell line. The antioxidant active extract showed significant stability against pH with attributed increase in antioxidant activity in the range between 8–12. The extract can be used effectively as a natural antioxidant and anticancer after progressive testing.

**Keywords:** *Nostoc linckia*; chemical compositions; oxidative stress; anticancer activity; antioxidant activity

## 1. Introduction

Cyanobacteria are prokaryote that have a simple cellular structure, and photosynthesis-like plants but don't have plant cell walls as with primitive bacteria. Cyanobacteria are considered as valuable sources of natural bioactive compounds, such as phenolic, flavonoids, terpenoids, proteins, lipids, vitamins, polysaccharides with considerable nutritional and medicinal value [1–5].

*Nostoc* genus is considered an edible source, and the genus belongs to the Nostocaceae family division of Cyanophyta. It forms spherical cells, which link together, forming filaments. Heterocysts design homogeneous cells that make up the filament and distance between vegetative cells [6,7]. Extracts from *Nostoc* biomass have been used in the medical field for fistula treatment, cancer therapy, anti-inflammatory, immune-boosting,

and blood pressure control [8]. Moreover, *Nostoc* sp. creates numerous compounds with an antimicrobial and antiviral activity, which significantly improved its cultivation [9]. A potential protein molecule, cyanovirin, generated by *Nostoc* sp., has been shown to obtain a pronounced effect in treatment of HIV and also influenza A virus [10–25]. *Nostoc* sp. includes polyunsaturated fatty acids that contain essential fatty acids, such as octade-catetraenoic, linoleic, α-linolenic, γ-linolenic acid, and eicosapentaenoic acid [26]. Many reports have been identified algal antioxidants such as glutathione, tocopherols and ascorbate [16–24]. Additionally, the algal extracts have proven antimicrobial, antiviral, and antitumor activities [25,26].

Different types of environmental stress (biotic and abiotic) lead to the disruption of homeostasis due to stressor application. As cells acclimate and strive to restore equilibrium, stress responses cause alterations in cell metabolism. Regulation, alarming stage, and adaptability are various steps of stress responses [27]. Stress strategies have been used to enhance high synthesis of active compounds using individual stress parameters, such as dietary factors (e.g., nitrogen, and carbon) and environmental factors (e.g., pH, temperature, light intensities, and salinity) [28–30]. Cu and Zn at high concentration has been shown to produce oxidative stress in an alga test by enhancing lipid peroxidation, membrane permeability, and lowering sulfhydryl level [30].

There is a lack of information regarding the influence of chemicals and biological responses manipulating conditions for growth. This study aimed to identify the impact of raising levels of copper and/or zinc in the *Nostoc linckia* culture medium (BG-$11_0$) on growth and secondary metabolite production and to assess the antioxidant and anticancer activity of the extracts.

## 2. Materials and Methods

### 2.1. Materials

#### 2.1.1. Chemicals and Reagents

Acetone, chloroform, diethyl ether, hexane, ether, ethanol, methylene chloride and methanol were provided from E. Merck Co. (Darmstadt, Germany). 2,2′-azino-bis (ethylben-zthiazoline-6-sulfonic acid (ABTS$^+$), sulfarhodamine, 2,2 diphenyl-1-picrylhydrazyl (DPPH), were provided from Sigma–Aldrich (St. Louis, MO, USA). Other ingredients, such as trichloroacetic acid, were of finest commercial grade obtainable. Sigma–Aldrich provided gallic acids, butylated hydroxyl toluene (BHT), rutin, tannic acid, L-proline, and vitamin C (St. Louis, MO, USA).

#### 2.1.2. Cell Line Cultures

In DMEM medium, a human lung cancer cell line (A549) was grown. Human breast adenocarcinoma (MCF-7), human colorectal cancer (HCT 116), and human cervical cancer (Hela) were all grown in RPMI-1640 medium with high glucose (DMEM high glucose w/steady glutamine w/sodium pyruvate, Biowest). L-glutamine and 10% fetal bovine serum (FBS) and 1% antibiotic (antibiotic antimycotic, Biowest, cat number) were added to both media. For growth, cells were incubated in 5% $CO_2$ humidified at 37 °C. Cell lines were produced and grown at Al Azhar University's Faculty of Medicine's, Center of Genetic Engineering.

### 2.2. Antioxidant Enzyme Kits

#### 2.2.1. Catalase Enzyme (Bio Diagnostic, Cairo, Egypt)

(1) Phosphate buffer pH7 (100 mM/L), (2) $H_2O_2$ substrate (500 mM/L), (3) chromogen inhibitor, and (4) enzyme peroxidase (>2000 U/L), 4-Aminoantipyrine preservative (2 mM/L) are included in the kit.

#### 2.2.2. Glutathione-S-Transferase (GST) (Bio Diagnostic, Cairo, Egypt)

By monitoring the absorbance at 340 nm, the GST kit determines total GST activity.

2.2.3. Lipid Peroxidation (Malondialdehyde) (Bio Diagnostic, Cairo, Egypt)

The kit used for MDA determination includes (1) standard (10 nmol/mL) and (2) thio-barbituric acid detergent stabilizer (25 mmol/L).

### 2.3. Methods

2.3.1. Cyanobacteria Cultivation

Dr. Sanaa Shanab, Professor of Phycology at Botany and Microbiology Department, Faculty of Science, Cairo University, kindly isolated and characterized cyanobacterium *Nostoc linckia* used for this study in the phycology lab [31,32].

As previously reported [4], *Nostoc linckia* was kept in standard conditions at $25 \pm 1\,°C$ in BG-110 medium, under fluorescent white light (Philips, TLD18W/54-765) of intensity $40\ \mu mol\ m^{-2}\ s^{-1}$, light duration (16-8 L/D cycles), and constant bubbling of air (filtered through a 0.22 μm microporous filter) with constant bubbling of air (filtered through a 0.22 m microporous filter).

2.3.2. Chemical Composition Modification of Culture Medium

An increase or reduction in certain element concentration (as single element stress) was undertaken to study the effect of various nutritive elements provided by a culture medium (BG-110), as shown in Table 1. The concentrations of copper and zinc utilized were zero, 0.158, 0.316, and 0.632 mg/L for copper, and zero, 0.44, 0.88, and 1.76 mg/L for zinc, respectively. Copper and zinc were utilized in combination at concentrations of 0.316 mg/L for copper and 0.88 mg/L for zinc (as double element stress).

**Table 1.** Algal culture medium (BG-11$_0$) and its modifications.

| Macronutrients | Control Culture Condition (g/500 mL) | Nutrients of Modified Culture with Copper (g/500 mL) | Nutrients of Modified Culture with Zinc (g/500 mL) |
|---|---|---|---|
| $NaHCO_3$ | 0.21 | 0.21 | 0.21 |
| $K_2HPO_4$ | 2.0 | 2.0 | 2.0 |
| $MgSO_4·7H_2O$ | 3.75 | 3.75 | 3.75 |
| $CaCl_2·2H_2O$ | 1.8 | 1.8 | 1.8 |
| Citric acid | 0.3 | 0.3 | 0.3 |
| Amm. Ferric citrate | 0.3 | 0.3 | 0.3 |
| $EDTANa_2$ | 0.05 | 0.05 | 0.05 |
| $Na_2CO_3$ | 1.0 | 1.0 | 1.0 |
| Then take 10 mL from each stock (per 1 L) | | | |
| Trace metals | g/L | g/L | g/L |
| $H_3BO_3$ | 2.86 | 2.86 | 2.86 |
| $MnCl_2·4H_2O$ | 1.81 | 1.81 | 1.81 |
| $ZnSO_4·7H_2O$ | 0.22 | 0.22 | 0, 0.44, 0.88 and 1.76 |
| $Na_2MoO_4·2H_2O$ | 0.39 | 0.39 | 0.39 |
| $CuSO_4·5H_2O$ | 0.08 | 0, 0.158, 0.316 and 0.632 | 0.08 |
| $Co (NO_3)_2·6H_2O$ | 0.05 | 0.05 | 0.05 |
| Then take 1 mL from trace elements stock (per 1 L) | | | |

2.3.3. Growth Rates Determination

The optical density at 550 nm was measured at 5-day intervals during a 30-day incubation period under different culture conditions to assess growth rate of *Nostoc linckia* axenic culture (free of bacteria).

### 2.3.4. Cyanobacteria Dry Weight

Dry biomass of cyanobacterium was estimated after filtering, washing, and drying at 105 °C for 24 h at 5-day intervals of incubation period (using 20 mL cyanobacteria suspension) [33].

### 2.3.5. Water-Soluble Pigments (Phycobiliproteins)

Crude extract was produced by suspending the biomass in a potassium phosphate buffer at pH 7 and ultrasonically disrupting cells, then centrifuging the extract to clarify it (10,000 rpm). Total phycobiliproteins were quantified using spectrophotometer and calculated in mg/g dry cell weight (DW) using following equations [34].

$$\text{Phycocyanin (PC)} = ((A615) - (0.474 \text{x } A652))/5.34$$

$$\text{Allophycocyanin (APC)} = ((A652) - (0.208 \text{x } A615))/5.09$$

$$\text{Phycoerythrin (PE)} = ((A562) - (2.41 \text{x PC}) - (0.849 \text{x APC}))/9.62$$

$$\text{Total Phycobiliproteins} = PC + PE + APC$$

where A615, A652 and A562 were the absorbance at 615, 652 and 562 nm, respectively.

### 2.3.6. Total Phenolic Contents

Phenolic contents in crude extract were measured using Taga et al. [35] method and represented as Gallic acid equivalent/gram of cyanobacteria (GAE/g).

### 2.3.7. Flavonoids Contents

The flavonoid amount in cyanobacterial extract was measured using Quettier et al. [36] and represented as rutin equivalent (mg of Ru/g of extract).

### 2.3.8. Tannins Contents

Using the vanillin hydrochloride method of Broadhurst and Jones [37], the amount of tannic acid in samples was analyzed from the standard curve and represented as tannic equivalents.

### 2.3.9. Aqueous Protein Determination

Aqueous protein concentration was estimated using Lowry et al. [38], and absorbance was measured at 720 nm, according to Rausch's slight modification [39]. The protein concentration was calculated from BSA calibration curve.

### *2.4. Antioxidant Enzymes and Lipid Peroxidation Determination*
### 2.4.1. Catalase Estimation

Cyanobacteria homogenate and phosphate buffer+$H_2O_2$ at 25 °C were incubated for 1 min then with chromogen –inhibitor, enzyme peroxidase and preservative after incubation for 10 min at 37 °C, recording the absorbance of sample against the sample blank and standard against standard blank utilizing methods of Aebi [40] and Fossati et al., [41], calculating activity as U/g = A standard − A sample/A standard × 1/gm biomass used.

### 2.4.2. Glutathione-S-Transferase

Cyanobacteria homogenate and phosphate buffer at pH 7.4 + glutathione reduced (GSH) incubation at 37 °C for 5 min, then insert CDNB and incubate at 37 °C for 5 min, then combine with TCA, centrifuge at 3000 rpm for 5 min, record the absorbance of sample (A sample) against a blank at 340 nm, calculate activity as U/g biomass = A sample × 2.812/g biomass sample [42].

### 2.4.3. Lipid Peroxidation (MDA Determination Method)

Cyanobacteria homogenate and phosphate buffer (pH 7.5) were centrifuged at 4000 rpm for 15 min. Then the supernatant and chromogen were heated in a boiling water bath for 30 min, cooled and the absorbance of the cyanobacteria sample were measured (A sample) against blank and standard against distilled water at 354 nm [43]. MDA in cyanobacteria sample (nmol/g cyanobacteria biomass) = A sample/A standard $\times$ 10/g tissue.

### 2.5. Biological Activities of Cyanobacteria Extract

Biological activities determination of crude extract obtained from *Nostoc linckia* cultivated in the $BG-11_0$ growth medium (control) and altered (stressed) medium $BG-11_0$ with altered contents of Zinc and Cu, single or in combination, the following activities were conducted.

### 2.5.1. Antioxidant Activity
### DPPH Scavenging Activity

Crude methanol's scavenging properties: a methylene chloride (1:1) extract was determined by adding 2.0 mL of 0.16 mM DPPH solution (in methanol) to test tube holding a 2.0 mL aliquot of material [44]. Mixture was vortexed for 1 min before being held at room temperature in the dark for 30 min. All sample solutions and BHT as a synthetic standard had their absorbance measured at 517 nm. The following formula was used to calculate % of scavenging activity:

$$\% \text{ Antioxidant activity} = (\text{Control-Sample} \times 100)/\text{Control}$$

where: control is DPPH solution (0.16 mM).

### ABTS Scavenging Assay

In comparison to a standard, the assay is based on the ability of different compounds to scavenge (2,2'-azino-bis ethylbenzthiazoline-6-sulfonic acid) ($ABTS^+$) radical cation (BHT). Radical cation was produced by combining 7 mM ABTS stock solution with 2.45 mM potassium persulfate (1/1, *v/v*) and allowing mixture to sit for 4–16 h until reaction was complete and the absorbance was steady. For measurements, $ABTS^+$ solution was diluted with ethanol to an absorbance of $0.700 \pm 0.05$ at 734 nm [45]. The photometric assay used 0.9 mL of $ABTS^+$ and 0.1 mL of tested substances, which were mixed for 45 s and measured at 734 nm after 1 min. Antioxidant activity of tested samples and standards (BHT and Vit C) was determined by calculating the decrease in absorbance at various concentrations using equation: $E = ((Ac - At)/Ac) \times 100$, where: At and Ac are respective absorbance of tested samples or standard and $ABTS^+$.

### 2.5.2. Anticancer Activity
### Cell Proliferation Evaluation of by MTT Assay

MTT (3-[4,5-methylthiazol-2-yl]-2,5-diphenyl-tetrazolium bromide) test was used to assess the anticancer impact of crude extracts and fractions of Nostoc linckia on different cancer cell lines, as previously reported [46]. In brief, cancer cells ($1 \times 10^4$ cells/well) were planted on triplicate in a 96-well plate and left to adhere for 24 h following cell count and vitality were determined using trypan blue dye.

Because the final concentration of DMSO in culture medium never exceeded 0.2% (*v/v*), the crude extracts were weighted and dissolved in 1 μml dimethyl sulfoxide (DMSO) to make a stock solution of 2000 μg/mL, and then different concentrations of crude extracts were equipped by diluting them in complete medium to make final concentrations of 31.5, 62.5, 125, 250, and 500 μg/mL [47]. The medium was replaced the next day with a new medium containing the stated amounts of the tested substances, and the cells were allowed to proliferate for 48 h. Each well received 10 μL of MTT (5 mg/mL in PBS w/o Ca, Mg, Lonza Verviers SPRL Belgium, cat number17-516F) 4 h before the incubation was

completed. After the incubation period was completed, 100 µL dimethyl sulfoxide was added to each well, and the 96 well plates were centrifuged for 5 min. at 4000 rpm to precipitate the formazan crystals. Using a Bio-Tek microplate reader, the color formed after reaction was quantified at 490 nm.

The experiment was repeated three times, and results were calculated as a percentage of cell viability using formula: % cell viability = (mean absorbance in test wells/mean absorbance in control wells) × 100.

### 2.6. Fractionation of Nostoc Linckia Promising Extract (0.44 mg/L Zn)

Hexane was used to pack 150 g silica gel (60–120 mesh for column chromatography) into a 40 cm long, 2.5 cm diameter chromatographic column. 3 g methylene chloride: methanol (1:1) crude extract of *Nostoc linckia* cultured under 0.44 mg/L Zn (as a promising crude extract) was finely mixed with some silica gel powder and then put on top of the packed column (Table 2). The column was then eluted successively with 100% hexane and raised polarity with chloroform and ethyl acetate, with polarity increasing by 10% between each mobile phase mixture (a total of 21 fractions were produced) as follows:

**Table 2.** Total fractions obtained from chromatographic column fractionation.

| Solvent | Fractions No. | | | | | | | | | | | | | | | | | | | | |
|---|---|---|---|---|---|---|---|---|---|---|---|---|---|---|---|---|---|---|---|---|---|
| | 1 | 2 | 3 | 4 | 5 | 6 | 7 | 8 | 9 | 10 | 11 | 12 | 13 | 14 | 15 | 16 | 17 | 18 | 19 | 20 | 21 |
| Hexane | 100 | 90 | 80 | 70 | 60 | 50 | 40 | 30 | 20 | 10 | 0 | 0 | 0 | 0 | 0 | 0 | 0 | 0 | 0 | 0 | 0 |
| Chlorofom | 0 | 10 | 20 | 30 | 40 | 50 | 60 | 70 | 80 | 90 | 100 | 100 | 90 | 80 | 70 | 60 | 50 | 40 | 30 | 20 | 10 |
| Ethyl acetate | 0 | 0 | 0 | 0 | 0 | 0 | 0 | 0 | 0 | 0 | 0 | 0 | 10 | 20 | 30 | 40 | 50 | 60 | 70 | 80 | 90 |

### 2.7. Gas Chromatography-Mass Spectrometry

Chemical composition was characterized using Trace GC1300-TSQ mass spectrometer (Thermo Scientific, Austin, TX, USA) with direct capillary column TG–5MS (30 m × 0.25 mm × 0.25 µm film thickness). The column oven temperature was initially held at 60 °C and then increased by 5 °C /min to 200 °C holds for 2 min increased to final temperature 300 °C by 20 °C /min and held for 2 min. The injector and MS transfer line temperatures were kept at 250 and 260 °C, respectively; helium was used as a carrier gas at a constant flow rate of 1 mL/min. The solvent delay time was 3 min, and diluted samples of 1 µL were injected automatically using autosampler AS1300 coupled with GC in the split mode. EI mass spectra were collected at 70 eV ionization voltages over the range of $m/z$ 50–650 in full scan mode. The ion source temperature was set at 250 °C. Components were identified by comparison of their retention times and mass spectra with those of WILEY 09 and NIST 11 mass spectral database.

### 2.8. pH Stability of the Extracts

The effect of pH values of the different solvent systems on the stability of algal extracts was carried out by the method reported by Majo et al., (2011) [48]. Briefly, phosphate buffer solutions of different pH values (3.0, 5.0, 7.0, 8.0, 10.0 and 12.0) were prepared with disodium hydrogen phosphate and citric acid, resulting in the preparation of 0.5 mg/mL sample solutions with different pH values. Then, the antioxidant activity of algal extract was determined using ABTS assay [49].

### 2.9. Statistical Analysis

The data were presented as mean ± standard deviation of three determinations. A one-way analysis of variance was used for statistical comparison, followed by Duncan's multiple range test (DMRT). Significant $p$-values of less than 0.05 ($p < 0.05$) were evaluated.

## 3. Results

### *3.1. Growth Rate*

Table 3 shows the growth rate of *Nostoc linckia* grown on BG-110 medium with various copper concentrations. Obtained data illustrated that the cyanobacteria cells significantly reached the maximum growth rate (as OD and Dwt.) on the twenty-fifth day of cultivation; after that, the growth rate declined until the 30th day of planting. Under copper deprivation conditions, a comparable growth rate was recorded ($0.6 \pm 0.026$, $0.88 \pm 0.010$ as Dwt and OD, respectively). Elevation of Cu concentration (0.158, 0.316, and 0.632 mg/L) significantly retarded growth rate gradually as a function of increasing metal concentration for the first 20 days. Growth was stopped until the end of the experiment.

Similarly, data in Table 4 show a noticeable decrease in growth rate during the incubation period from the zero time to the twentieth day of cultivation due to increasing Zn concentration (0.0–0.632 mg/L) in the cultivation media. Obtained data showed that the absence of zinc in cultivation media enhanced the growth rate gradually during the first 25 days of cultivation period with no significant differences ($0.72 \pm 0.021$, $0.83 \pm 0.006$ and $0.79 \pm 0.014$, $0.88 \pm 0.010$ in Dwt and OD, respectively) compared with control (0.22 mg/L Zn). At higher Zn concentration (0.44, 0.88 and 1.76 mg/L) growth was significantly decreased (by around 50%) compared to control treatment and completely inhibited after twenty days of cultivation.

Table 5 shows the growth rate of *Nostoc linckia* when combining the two micronutrients Cu and Zn (0.316 mg/L Cu and 0.88 mg/L Zn). A significant decrease in growth rate was recorded after 15 days of cultivation compared with control treatment and complete growth inhibition was noticed after 20 days.

**Table 3.** Growth rate of *Nostoc linckia* (as O.D and Dwt). (g/L) grown under various concentrations of copper ion.

| Days | Cu Deprivation (Cu) | | Control (0.079 mg/L) | | 1× Cu (0.158 mg/L) | | 2× Cu (0.316 mg/L) | | 4× Cu (0.632 mg/L) | |
|---|---|---|---|---|---|---|---|---|---|---|
| | Dwt. | O.D | Dwt. | O.D | Dwt. | O.D | Dwt. | O.D | Dwt. | O.D |
| 0 | 0.05 ± 0.005 [f] | 0.01 ± 0.002 [g] | 0.05 ± 0.005 [g] | 0.01 ± 0.002 [g] | 0.05 ± 0.002 [d] | 0.01 ± 0.003 [d] | 0.04 ± 0.005 [c] | 0.01 ± 0.002 [e] | 0.05 ± 0.005 [b] | 0.01 ± 0.002 [d] |
| 5 | 0.08 ± 0.005 [f] | 0.06 ± 0.005 [f] | 0.11 ± 0.010 [f] | 0.07 ± 0.005 [f] | 0.07 ± 0.006 [c] | 0.07 ± 0.006 [c] | 0.06 ± 0.010 [c] | 0.06 ± 0.002 [d] | 0.06 ± 0.010 [b] | 0.04 ± 0.004 [c] |
| 10 | 0.18 ± 0.010 [e] | 0.30 ± 0.005 [e] | 0.21 ± 0.012 [e] | 0.38 ± 0.004 [e] | 0.18 ± 0.005 [b] | 0.17 ± 0.005 [b] | 0.13 ± 0.006 [b] | 0.12 ± 0.006 [c] | 0.10 ± 0.010 [a] | 0.09 ± 0.005 [a] |
| 15 | 0.28 ± 0.008 [d] | 0.42 ± 0.007 [d] | 0.31 ± 0.007 [d] | 0.46 ± 0.007 [d] | 0.29 ± 0.010 [a] | 0.29 ± 0.008 [a] | 0.26 ± 0.010 [a] | 0.25 ± 0.008 [a] | 0.11 ± 0.005 [a] | 0.10 ± 0.005 [a] |
| 20 | 0.39 ± 0.005 [c] | 0.49 ± 0.005 [c] | 0.43 ± 0.007 [c] | 0.54 ± 0.006 [c] | 0.18 ± 0.006 [b] | 0.16 ± 0.008 [b] | 0.15 ± 0.010 [b] | 0.14 ± 0.005 [b] | 0.06 ± 0.006 [b] | 0.05 ± 0.006 [b] |
| 25 | 0.60 ± 0.026 [a] | 0.80 ± 0.015 [a] | 0.79 ± 0.014 [a] | 0.88 ± 0.010 [a] | _ | _ | _ | _ | _ | _ |
| 30 | 0.51 ± 0.010 [b] | 0.59 ± 0.005 [b] | 0.67 ± 0.008 [b] | 0.71 ± 0.015 [b] | _ | _ | _ | _ | _ | _ |

In each column, data are provided as means ± SD (n = 3), and means with various letters are significantly different ($p < 0.05$) for each concentration.

**Table 4.** Growth rate of *Nostoc linckia* (as O.D and Dwt). (g/L) grown under various concentrations of zinc ion.

| Days | Zn Deprivation | | Control (0.22 mg/L) | | 1× Zn (0.44 mg/L) | | 2× Zn (0.88 mg/L) | | 4× Zn (1.76 mg/L) | |
|---|---|---|---|---|---|---|---|---|---|---|
| | Dwt. | O.D | Dwt. | O.D | Dwt. | O.D | Dwt. | O.D | Dwt. | O.D |
| 0 | 0.05 ± 0.005 [g] | 0.01 ± 0.002 [g] | 0.05 ± 0.005 [g] | 0.01 ± 0.002 [g] | 0.05 ± 0.002 [d] | 0.01 ± 0.003 [e] | 0.04 ± 0.005 [d] | 0.01 ± 0.002 [d] | 0.05 ± 0.005 [d] | 0.01 ± 0.002 [e] |
| 5 | 0.09 ± 0.005 [f] | 0.28 ± 0.005 [f] | 0.11 ± 0.010 [f] | 0.07 ± 0.005 [f] | 0.07 ± 0.002 [c] | 0.06 ± 0.005 [d] | 0.07 ± 0.006 [c] | 0.05 ± 0.004 [c] | 0.05 ± 0.007 [d] | 0.04 ± 0.004 [d] |
| 10 | 0.19 ± 0.008 [e] | 0.31 ± 0.005 [e] | 0.21 ± 0.012 [e] | 0.38 ± 0.004 [e] | 0.19 ± 0.006 [b] | 0.20 ± 0.007 [b] | 0.19 ± 0.005 [b] | 0.13 ± 0.007 [b] | 0.16 ± 0.005 [b] | 0.10 ± 0.005 [b] |
| 15 | 0.29 ± 0.005 [d] | 0.44 ± 0.006 [d] | 0.31 ± 0.007 [d] | 0.46 ± 0.007 [d] | 0.30 ± 0.002 [a] | 0.33 ± 0.006 [a] | 0.28 ± 0.008 [a] | 0.28 ± 0.007 [a] | 0.25 ± 0.007 [a] | 0.12 ± 0.007 [a] |
| 20 | 0.40 ± 0.007 [c] | 0.50 ± 0.005 [c] | 0.43 ± 0.007 [c] | 0.54 ± 0.006 [c] | 0.21 ± 0.005 [b] | 0.18 ± 0.010 [c] | 0.18 ± 0.005 [b] | 0.14 ± 0.005 [b] | 0.14 ± 0.005 [c] | 0.08 ± 0.005 [c] |
| 25 | 0.72 ± 0.021 [a] | 0.83 ± 0.006 [a] | 0.79 ± 0.014 [a] | 0.88 ± 0.010 [a] | _ | _ | _ | _ | _ | _ |
| 30 | 0.55 ± 0.01 [b] | 0.61 ± 0.010 [b] | 0.67 ± 0.008 [b] | 0.71 ± 0.015 [b] | _ | _ | _ | _ | _ | _ |

In each column, data are provided as means ± SD (n = 3), and means with various letters are significantly different ($p < 0.05$) for each concentration.

**Table 5.** Growth rate of *Nostoc linckia* (as O.D and Dwt). (g/L) grown on BG-110 medium with adjusted Cu and Zn ion concentrations and combined impact of two elements.

| Days | Control Condition (0.079 mg/L Cu + 0.22 mg/L Zn) | | Stressed Condition (0.316 mg/L Cu + 0.88 mg/L Zn) | |
| --- | --- | --- | --- | --- |
| | **Dwt.** | **O.D** | **Dwt.** | **O.D** |
| 0 | $0.05 \pm 0.005$ [g] | $0.01 \pm 0.002$ [g] | $0.05 \pm 0.005$ [d] | $0.01 \pm 0.002$ [d] |
| 5 | $0.11 \pm 0.010$ [f] | $0.07 \pm 0.005$ [f] | $0.07 \pm 0.007$ [c] | $0.04 \pm 0.007$ [c] |
| 10 | $0.21 \pm 0.012$ [e] | $0.38 \pm 0.004$ [e] | $0.11 \pm 0.008$ [b] | $0.09 \pm 0.005$ [b] |
| 15 | $0.31 \pm 0.007$ [d] | $0.46 \pm 0.007$ [d] | $0.20 \pm 0.005$ [a] | $0.25 \pm 0.009$ [a] |
| 20 | $0.43 \pm 0.007$ [c] | $0.54 \pm 0.006$ [c] | $0.07 \pm 0.007$ [c] | $0.06 \pm 0.007$ [c] |
| 25 | $0.79 \pm 0.014$ [a] | $0.88 \pm 0.010$ [a] | – | – |
| 30 | $0.67 \pm 0.008$ [b] | $0.71 \pm 0.015$ [b] | – | – |

In each column, data are provided as means $\pm$ SD (n = 3), and means with various letters are significantly different ($p < 0.05$) for each concentration.

### 3.2. Phycobiliprotein Content

The data in Table 6 reveal that at the lowest copper concentration, the maximum phycocyanin pigment content ($60.64 \pm 3.42$ µg/g) and phycoerythrin pigments ($123.05 \pm 4.59$ µg/g) had been achieved. In contrast, the maximum allophycocyanin content was produced at 0.316 mg/L Cu. Significantly high phycocyanin and phycoerythrin pigments were produced ($60.64 \pm 3.42$ and $123.05 \pm 4.59$ µg/g, respectively) at zinc starvation. On the contrary, the lowest production of total phycobiliprotein ($10.47 \pm 1.13$ µg/g) was measured at the elevated zinc concentration (1.76 mg/L). A combination of Cu and Zn resulted in significant elevation in total phycobiliprotein ($119.76 \pm 6.17$ µg/g) compared with the control treatment (0.079 mg/L of Cu and 0.22 mg/L of Zn). A non-significant difference in total phycobiliprotein content was noticed in copper deprivation treatment and combined Cu–Zn treatment. On the other hand, Zn starvation showed significant high content of total phycobiliprotein ($200.49 \pm 5.36$ µg/g) compared with all treatments.

**Table 6.** Phycobiliprotein content (as µg/g) for *Nostoc linckia* grown on various copper and zinc concentrations.

| Treatments | Phycocyanin | Allophycocyanin | Phycoerythrin | Total Phycobiliprotein |
| --- | --- | --- | --- | --- |
| **Zinc concentration (mg/L)** | | | | |
| 0 | $60.64 \pm 3.42$ [a] | $17.80 \pm 2.15$ [b] | $123.05 \pm 4.59$ [a] | $200.49 \pm 5.36$ [a] |
| 0.22 | $51.09 \pm 3.55$ [b] | $25.96 \pm 2.12$ [a] | $82.71 \pm 4.10$ [b] | $159.76 \pm 6.17$ [b] |
| 0.44 | $31.26 \pm 2.45$ [d] | $14.55 \pm 1.104$ [c] | $48.30 \pm 3.37$ [d] | $94.11 \pm 5.44$ [d] |
| 0.88 | $13.53 \pm 1.34$ [e] | $0.94 \pm 0.005$ [d] | $30.42 \pm 3.50$ [e] | $44.90 \pm 2.15$ [e] |
| 1.76 | $3.88 \pm 1.01$ [f] | $0.064 \pm 0.003$ [e] | $6.52 \pm 1.08$ [f] | $10.47 \pm 1.13$ [f] |
| **Copper concentration (mg/L)** | | | | |
| 0 | $38.81 \pm 2.06$ [c] | $20.34 \pm 2.28$ [b] | $58.12 \pm 3.05$ [c] | $116.27 \pm 6.48$ [c] |
| 0.079 | $60.64 \pm 3.42$ [a] | $17.80 \pm 2.15$ [c] | $123.05 \pm 4.59$ [a] | $200.49 \pm 5.36$ [a] |
| 0.158 | $41.50 \pm 3.44$ [b] | $24.34 \pm 2.41$ [a] | $69.18 \pm 4.50$ [b] | $135.02 \pm 5.50$ [b] |
| 0.316 | $36.22 \pm 3.60$ [d] | $26.43 \pm 2.30$ [a] | $58.57 \pm 4.32$ [c] | $121.23 \pm 5.46$ [c] |
| 0.632 | $23.21 \pm 2.48$ [e] | $4.946 \pm 0.57$ [d] | $32.20 \pm 1.52$ [d] | $60.35 \pm 4.45$ [d] |
| 0.88 mg/L Zn + 0.316 mg/L Cu | $41.09 \pm 2.50$ [b] | $15.96 \pm 1.02$ [c] | $62.73 \pm 4.10$ [b] | $119.76 \pm 6.17$ [c] |

In each column, data are provided as means $\pm$ SD (n = 3), and means with various letters are significantly different ($p < 0.05$) for each concentration.

### 3.3. Phenolic, Flavonoids and Tannins Content

Data in Table 7 demonstrate the content of secondary metabolites at different levels of Zn and Cu ions in *Nostoc linckia* culture media. The data revealed that cultivation of cyanobacteria cells at 0.316 mg/L of Cu produced a significant large content of phenolic and flavonoid content ($41.39 \pm 0.56$ mg/g and $20.10 \pm 0.85$ mg/g, respectively). On the other hand, zinc concentration at 0.44 mg/L induced the highest phenolic and flavonoids contents ($89.29 \pm 1.46$ mg/g and $21.05 \pm 0.92$ mg/g, respectively). Furthermore, combin-

ing 0.88 mg/L Zn with 0.316 mg/L Cu resulted in significantly decreased phenolic and flavonoid contents (15.84 ± 0.55 mg/g and 16.43 ± 0.51 mg/g, respectively). Regarding tannins content, our obtained data revealed that the elevated zinc concentration (1.76 mg/L) generated maximum tannins content (13.17 ± 0.37 mg/g). While at moderately elevated Cu concentration (0.158 mg/L), the highest tannins content (8.06 ± 0.82 mg/g) was measured. The amount of tannins created by combining 0.88 mg/L (Zn) and 0.316 mg/L (Cu) was enhanced to 9.15 ± 0.30 mg/g, which was higher than those produced independently by Cu and Zn at the same concentration.

**Table 7.** Phenolic, flavonoids and tannins content of *Nostoc linckia* cultured on different concentration of zinc and copper (and in combination).

| Treatments | Phenolics (as mg Gallic Acid Equivalent/g Dry wt.) | Flavonoids (as mg of Rutin/g Dry wt.) | Tannins (as mg Tannic Acid Equivalent/g Dry wt.) |
|---|---|---|---|
| | **Zinc conc. (mg/L)** | | |
| 0 | 14.80 ± 0.75 [c] | 7.08 ± 0.87 [d] | 10.99 ± 0.51 [b] |
| 0.22 | 24.25 ± 1.30 [b] | 18.13 ± 1.02 [b] | 5.45 ± 0.47 [d] |
| 0.44 | 89.29 ± 1.46 [a] | 21.05 ± 0.92 [a] | 0.524 ± 0.06 [f] |
| 0.88 | 17.27 ± 0.64 [c] | 10.74 ± 0.67 [c] | 2.75 ± 0.23 [e] |
| 1.76 | 17.01 ± 0.98 [c] | 9.56 ± 0.72 [c] | 13.17 ± 0.37 [a] |
| | **Copper conc. (mg/L)** | | |
| 0 | 23.98 ± 0.53 [d] | 6.17 ± 0.76 [c] | 5.27 ± 0.64 [b] |
| 0.079 | 24.25 ± 1.39 [d] | 18.13 ± 1.02 [ab] | 5.45 ± 0.47 [b] |
| 0.158 | 30.17 ± 0.75 [b] | 16.29 ± 0.62 [b] | 8.06 ± 0.82 [a] |
| 0.316 | 41.39 ± 0.56 [a] | 20.10 ± 0.85 [a] | 4.07 ± 0.99 [bc] |
| 0.632 | 26.86 ± 0.71 [c] | 19.80 ± 1.31 [a] | 3.12 ± 0.63 [c] |
| 0.88mg/L Zn+ 0.316mg/L Cu | 15.84 ± 0.55 [e] | 16.43 ± 0.51 [b] | 9.15 ± 0.30 [a] |

In each column, data are provided as means ± SD (n = 3), and means with various letters are significantly different ($p < 0.05$) for each concentration.

### 3.4. Antioxidant Activity

Table 8 reveals the antioxidant scavenging activity against DPPH and ABTS. In the current study, zinc treatment recorded the maximum antioxidant activity (92.36 ± 0.55, 94.22 ± 0.78% at 30 and 60 min, respectively for DPPH and 81.22 ± 0.69% for ABTS) at 0.44 mg/L of zinc. The lower Cu concentration (0.079 mg/L) showed significant high antioxidant activity against DPPH (91.25 ± 1.09 and 94.26 ± 0.80% at 30 and 60 min, respectively), which were even higher than those of synthetic standard BHT (85.63 ± 0.55 and 89.5 ± 0.55%, respectively). At the same copper concentration, the maximum antioxidant activity by ABTS assay was recorded (60.09 ± 1.00%).

When compared to BHT or control treatments, a combination of 0.88 mg/L Zn and 0.316 mg/L Cu recorded a significant decrease in antioxidant scavenging by DPPH (77.100.84% and 72.840.45% at 30 and 60 min., respectively). In contrast, a significant increase in activity was measured by ABTS assay (71.73 ± 0.30%) compared to the activity of separate elements (Zn or Cu) at the same concentration. Data in Table 9 show that the $IC_{50}$ of the promising extract (0.44 mg/L Zn) of *Nostoc linckia*, $IC_{50}$ were 13.0 and 13.1 µg/mL against DPPH assay at 30 and 60 min, respectively, compared to each antioxidant standards; BHT (11.1 and 11.2 µg/mL) and vitamin C (12.1 and 12.9 µg/mL), respectively at 30 and 60 min. While the $IC_{50}$ of the crude extract against ABTS was 16.1 µg/mL, near to synthetic (BHT) and vitamin C (15.1 and 14.7 µg/mL, respectively).

The antioxidant activity of *Nostoc linckia* extract (0.44 mg/L of Zn) screened on pH values ranging from 3–12 (Figure 1). Significant increase in ABTS scavenging activity was recorded at pH 8 and 10 (95.71 ± 0.41, 94.23 ± 0.46%, respectively) while the activity was reduced at pH values of 7 and 12 (89.22 ± 0.68 and 89.25 ± 1.53%, respectively). At the acidic pH (3 and 5), a significant decrease in antioxidant activity was noticed (51.15 ± 0.30 and 52.98 ± 0.13%, respectively).

**Table 8.** Antioxidant activity (%) of *Nostoc linckia* extract grown on various zinc and copper concentration against DPPH and ABTS radical methods.

| Treatments | DPPH | | ABTS |
|---|---|---|---|
| | 30 min | 60 min | |
| **Zinc conc. (mg/L)** | | | |
| 0 | 69.1 ± 1.01 [c] | 74.96 ± 0.95 [c] | 38.76 ± 1.15 [d] |
| 0.22 | 90.25 ± 1.09 [b] | 92.26 ± 0.80 [b] | 61.0 ± 1.00 [c] |
| 0.44 | 92.36 ± 0.55 [a] | 94.22 ± 0.78 [a] | 81.22 ± 0.69 [a] |
| 0.88 | 89.63 ± 0.40 [b] | 92.03 ± 0.84 [b] | 69.513 ± 0.84 [b] |
| 1.76 | 41.51 ± 0.94 [d] | 37.98 ± 0.22 [d] | 64.1 ± 0.95 [c] |
| **Copper conc. (mg/L)** | | | |
| 0 | 49.10 ± 0.85 [f] | 53.90 ± 1.01 [e] | 60.36 ± 0.55 [c] |
| 0.079 | 91.25 ± 1.09 [a] | 94.26 ± 0.80 [a] | 60.09 ± 1.00 [c] |
| 0.158 | 84.37 ± 0.54 [b] | 90.06 ± 0.20 [b] | 47.2 ± 0.43 [d] |
| 0.316 | 75.19 ± 0.73 [d] | 77.24 ± 0.76 [c] | 38.18 ± 0.74 [e] |
| 0.632 | 67.18 ± 0.93 [e] | 72.68 ± 0.47 [d] | 24.01 ± 0.88 [f] |
| 0.88mg/L Zn+ 0.316mg/L Cu | 77.1 ± 0.84 [c] | 72.84 ± 0.45 [d] | 71.73 ± 0.30 [b] |
| BHT | 85.63 ± 0.55 [b] | 89.50 ± 0.55 [b] | 90.82 ± 0.68 [a] |

In each column, data are provided as means ± SD (n = 3), and means with various letters are significantly different ($p < 0.05$) for each concentration.

**Table 9.** Antioxidant activity (as % and $IC_{50}$) of the effective extracts (0.44 mg/L (Zn) of *Nostoc linckia* compared with BHT and vitamin C as standards (using DPPH and ABTS radical methods).

| Extract Concentration (µg/mL) | Cruse Extract of (0.44 mg/L (Zn) | | | BHT | | | Vitamin C | | |
|---|---|---|---|---|---|---|---|---|---|
| | DPPH | | ABTS | DPPH | | ABTS | DPPH | | ABTS |
| | 30 min | 60 min | | 30 min | 60 min | | 30 min | 60 min | |
| 500 | 93.4 ± 1.0 [b] | 94.8 ± 2.3 [b] | 91.6 ± 2.0 [c] | 96.3 ± 1.5 [a] | 97.4 ± 3.0 [a] | 94.3 ± 3.0 [b] | 93.4 ± 1.1 [b] | 94.5 ± 2.8 [b] | 93.2 ± 2.5 [a] |
| 250 | 87.8 ± 0.7 [c] | 89.0 ± 1.9 [c] | 87.6 ± 2.0 [c] | 92.7 ± 1.0 [a] | 93.2 ± 2.2 [a] | 90.8 ± 2.9 [b] | 90.3 ± 1.0 [b] | 90.9 ± 2.2 [b] | 90.2 ± 2.1 [a] |
| 125 | 80.2 ± 0.6 [c] | 80.9 ± 1.3 [c] | 77.3 ± 1.9 [c] | 90.3 ± 0.9 [a] | 89.0 ± 2.0 [a] | 85.4 ± 2.0 [b] | 88.8 ± 0.9 [b] | 88.3 ± 1.9 [b] | 87.9 ± 1.9 [a] |
| 62.50 | 66.2 ± 0.7 [d] | 65.0 ± 1.0 [c] | 62.2 ± 1.3 [c] | 84.4 ± 0.4 [a] | 84.9 ± 2.0 [a] | 70.7 ± 1.0 [b] | 83.7 ± 1.1 [b] | 85.9 ± 1.0 [a] | 97.1 ± 3.6 [a] |
| 31.25 | 62.0 ± 0.4 [d] | 60.3 ± 1.1 [d] | 57.2 ± 0.9 [b] | 72.7 ± 0.5 [a] | 73.2 ± 1.9 [a] | 57.5 ± 0.9 [b] | 68.2 ± 0.7 [b] | 69.5 ± 0.9 [b] | 76.9 ± 1.9 [a] |
| 15.62 | 60.1 ± 0.9 [c] | 59.5 ± 0.8 [c] | 48.6 ± 0.8 [c] | 70.2 ± 0.8 [a] | 69.4 ± 1.2 [a] | 51.8 ± 0.9 [b] | 64.4 ± 0.5 [b] | 63.0 ± 0.9 [b] | 53.0 ± 1.0 [a] |
| $IC_{50}$ (ug/mL) | 13.0 | 13.1 | 16.1 | 11.1 | 11.2 | 15.1 | 12.1 | 12.9 | 14.7 |

In each column, data are provided as means ± SD (n = 3), and means with various letters are significantly different ($p < 0.05$) for each concentration.

Table 10 illustrates the main components of the extract of *Nostoc linckia* (0.44 mg/L Zn), which were screened by GC-mass spectroscopy. The analysis identified the following major active ingredients: cyclononasiloxane, hexadecenoic acid, 1H-purine-6-amine, 2 fluorophenyl methyl, 9,12,15 octadecatrienoic acid methyl ester and cyclodecasiloxane.

The data in Table 11 reveals that elevation of Zn concentration in growing media induced a gradual increase in lipid peroxidation. At the same time, 1.76 mg/L of zinc caused the maximum lipid peroxidation (1201.78 ± 20.33 nmol/g). Additionally, copper at a level of 0.632 mg/L rendered unique lipid peroxidation content (419.12 ± 10.43 nmol/g). The use of a combination of 0.88 mg/L Zn and 0.316 mg/L Cu resulted in significantly increased lipid peroxidation (929.93 ± 18.106 nmol/g) than those produced by individual Cu or Zn at the same concentrations (192.98 ± 8.42 and 683.25 ± 16.66 nmol/g). Glutathione-S-transferase (U/g tissue) activity was (335.10 ± 7.36 U/g) for the treatment with Zn at 1.76 mg/L, while the activity was 210.38 ± 7.74 U/g with Cu at 0.632 mg/L. However, a combination of 0.88 mg/L Zn with 0.316 mg/L Cu caused reduction in enzyme activity (165.94 ± 6.49U/g).

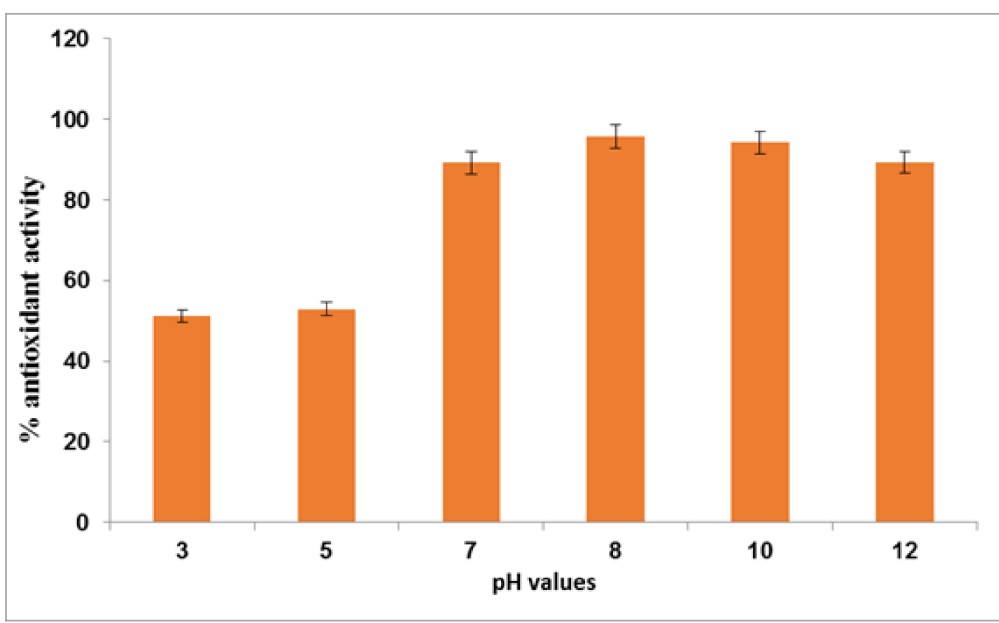

**Figure 1.** Antioxidant activity *Nostoc linckia* extract (0.44 mg/L of Zn) against ABTS radical at pH values 3,5,7,8,10 and 12.

**Table 10.** GC/MS analysis for most effective extract (0.44 mg/L Zn) of *Nostoc linckia* (as antioxidant).

| No. | Compound Name | (Relative Percentage) | Biological Activity | References |
|---|---|---|---|---|
| 1 | cyclononasiloxane | 9.73 | Antioxidant activity | Kadri et al., 2011 [50], Mebude et al., 2017 [51], Patil and Jadhav, 2014 [52] |
| 2 | 2-hexadecanol | 0.99 | Anti-inflammatory, antioxidant, and antimicrobial, anticancer activities | Kalaisezhiyen et al., 2012 [53] |
| 3 | hexadecanoic acid, methyl ester | 16.63 | Antioxidant, antimicrobial, hypocholesterolemia, nematicidal, pesticidal, hemolytic, antiandrogenic, hemolytic, 5-alpha reductase inhibitor cancer enzyme inhibitors in pharmaceutical, cosmetics, and food industries | Sayik et al., 2017 [54], Kalaisezhiyen et al., 2012 [53] |
| 4 | 2-hexadecen-1-ol, 3,7,11,15-tetramethyl | 1.17 | Anticancer, anti-inflammatory and antimicrobial, antioxidant activities | Kalaisezhiyen et al., 2012 [53], Mohamed et al., 2015 [55] |
| 5 | 7,9-di-tert-butyl-1-oxaspiro (4,5) deca-6,9-diene-2,8-dione | 0.78 | Antimicrobial activity | Aly et al., 2016 [56] |
| 6 | 1H-purin-6-amine, [(2-fluorophenyl)methyl] | 12.51 | Antimicrobial, antioxidant, cancer enzyme inhibitors in pharmaceutical, cosmetics, and food industries | Sayik et al., 2017 [54] |
| 7 | 9,12,15-octadecatrienoic acid, methyl ester | 3.67 | Antimicrobial, antioxidant, cancer enzyme inhibitors in pharmaceutical, cosmetics, and food industries | Prakash and Suneetha, 2014 [57], Sayik et al., 2017 [54] |
| 8 | Cyclodecasiloxane | 17.06 | Antibacterial activity | Jasim et al., 2015 [58] |
| 9 | 1-Eicosanol | 1.17 | - | - |
| 10 | 17-Pentatriacontene | 0.61 | - | - |
| 11 | Neophytadiene | 0.83 | Antioxidant, antibacterial activity | Swamy et al., 2017 [59] |
| 12 | 1-chlorooctadecane | 0.56 | - | - |

**Table 11.** Lipid peroxidation (MAD nmol/g), protein (mg/g), glutathione-S-transferase (U/g tissue), and catalase activity (%) generated by *Nostoc linckia* grown under various copper and zinc concentrations (separately or in combination).

| Treatments | Lipid Peroxidation (MAD n mol/g) | Protein (as mg/g) | GST (U/g) | Catalase Activity (%) |
|---|---|---|---|---|
| **Zinc conc. (mg/L)** | | | | |
| 0 | 145.09 ± 6.66 [e] | 221.06 ± 0.42 [b] | 89.78 ± 3.54 [e] | 40.00 ± 3.79 [d] |
| 0.22 | 109.67 ± 4.91 [f] | 252.58 ± 1.23 [a] | 81.04 ± 3.73 [f] | 40.90 ± 2.50 [d] |
| 0.44 | 583.42 ± 15.41 [d] | 136.70 ± 1.86 [c] | 188.19 ± 5.82 [c] | 41.19 ± 2.43 [d] |
| 0.88 | 683.25 ± 16.66 [c] | 52.81 ± 1.03 [d] | 260.85 ± 5.52 [b] | 46.92 ± 3.61 [c] |
| 1.76 | 1201.78 ± 20.33 [a] | 25.58 ± 1.23 [f] | 335.10 ± 7.36 [a] | 85.67 ± 3.29 [a] |
| **Copper conc. (mg/L)** | | | | |
| 0 | 128.036 ± 4.64 [e] | 208.40 ± 0.55 [b] | 102.73 ± 3.30 [e] | 47.99 ± 3.40 [c] |
| 0.079 | 109.67 ± 4.91 [f] | 252.58 ± 1.23 [a] | 81.04 ± 3.73 [f] | 40.90 ± 2.50 [d] |
| 0.158 | 174.02 ± 6.47 [d] | 127.68 ± 2.00 [c] | 126.81 ± 4.59 [d] | 58.02 ± 3.35 [a] |
| 0.316 | 192.98 ± 8.42 [c] | 115.83 ± 2.32 [d] | 172.38 ± 5.54 [b] | 50.78 ± 3.29 [b] |
| 0.632 | 419.12 ± 10.43 [b] | 68.04 ± 2.39 [e] | 210.38 ± 7.74 [a] | 52.61 ± 2.38 [b] |
| 0.88mg/L Zn + 0.316mg/L Cu | 929.93 ± 18.106 [a] | 41.97 ± 0.60 [f] | 165.94 ± 6.49 [c] | 50.01 ± 2.58 [b] |

In each column, data are provided as means ± SD (n = 3), and means with various letters are significantly different ($p < 0.05$) for each concentration.

Furthermore, we recorded the most significant catalase activity (85.67 ± 3.29%) in zinc treatment 1.76 mg/L. Similarly, copper at the highest treatment (0.632 mg/L) induced catalase activity of 52.61 ± 2.34%.

The promising crude extract (0.44 mg/L of Zn) was subjected for sub-fractionation. Sub-fractions had been tested for antioxidant activity by ABTS assay as shown in Table 12. The fraction numbers 1, 6, 7, and 8 were of higher antioxidant activity (74.35 ± 2.03, 56.41 ± 1.50, 53.80 ± 1.90, and 64.61 ± 2.05%, respectively, at 200 μg/mL) among of 21 fractions.

**Table 12.** Antioxidant activity (%) against ABTS assay of 21 fractions of the potential extract (0.44 mg/L Zn) of *Nostoc linckia* at 200 ppm.

| Fractions No. | Antioxidant Activity (%) |
|---|---|
| 1 | 74.35 ± 2.03 [a] |
| 2 | 27.17 ± 0.90 [f] |
| 3 | 28.97 ± 1.01 [f] |
| 4 | 32.82 ± 1.10 [e] |
| 5 | 33.84 ± 1.00 [e] |
| 6 | 56.41 ± 1.50 [c] |
| 7 | 53.80 ± 1.90 [c] |
| 8 | 64.61 ± 2.05 [b] |
| 9 | 36.66 ± 1.70 [d] |
| 10 | 32.30 ± 1.23 [e] |
| 11 | 32.25 ± 1.50 [e] |
| 12 | 36.15 ± 1.20 [d] |
| 13 | 33.33 ± 1.43 [e] |
| 14 | 28.46 ± 1.20 [f] |
| 15 | 28.00 ± 1.06 [f] |
| 16 | 23.58 ± 1.05 [g] |
| 17 | 25.89 ± 1.09 [g] |
| 18 | 27.02 ± 0.98 [f] |
| 19 | 32.30 ± 1.11 [e] |
| 20 | 31.79 ± 1.90 [e] |
| 21 | 38.46 ± 1.77 [d] |

For each extract, different small letters in column represent a significant difference ($p < 0.05$). Error bars indicate ± SD of three replicates.

### 3.5. Antitumor Activity

Figure 2 shows the percentage of cell viability of four cancer cell lines examined, whereas Figure 3 shows $IC_{50}$. The crude extract (0.44 mg/L Zn) had a strong potency towards cyanobacteria in HeLa and MCF-7 cell lines, with $IC_{50}$ of 65.8 and 66.7 μg/mL, respectively, followed by A549 and HCT 116 cells, with $IC_{50}$ of 98.3 and 204 μg/mL, respectively.

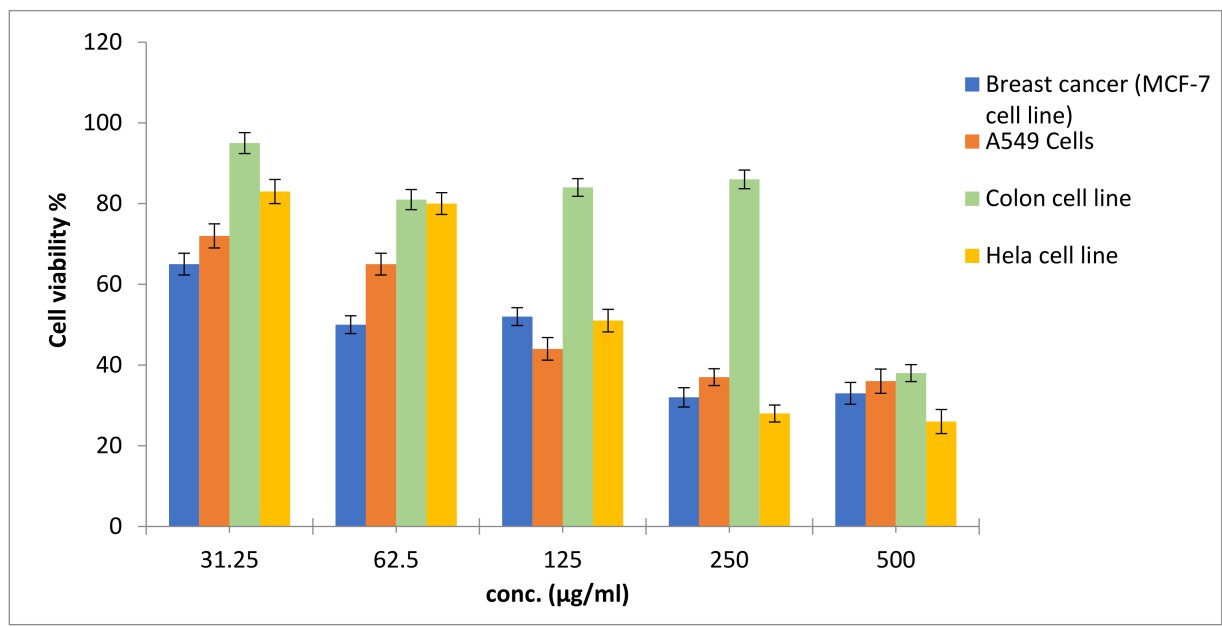

**Figure 2.** Anticancer activity of the tested extract (0.44 mg/L Zn of *Nostoc linckia*) against four cell lines.

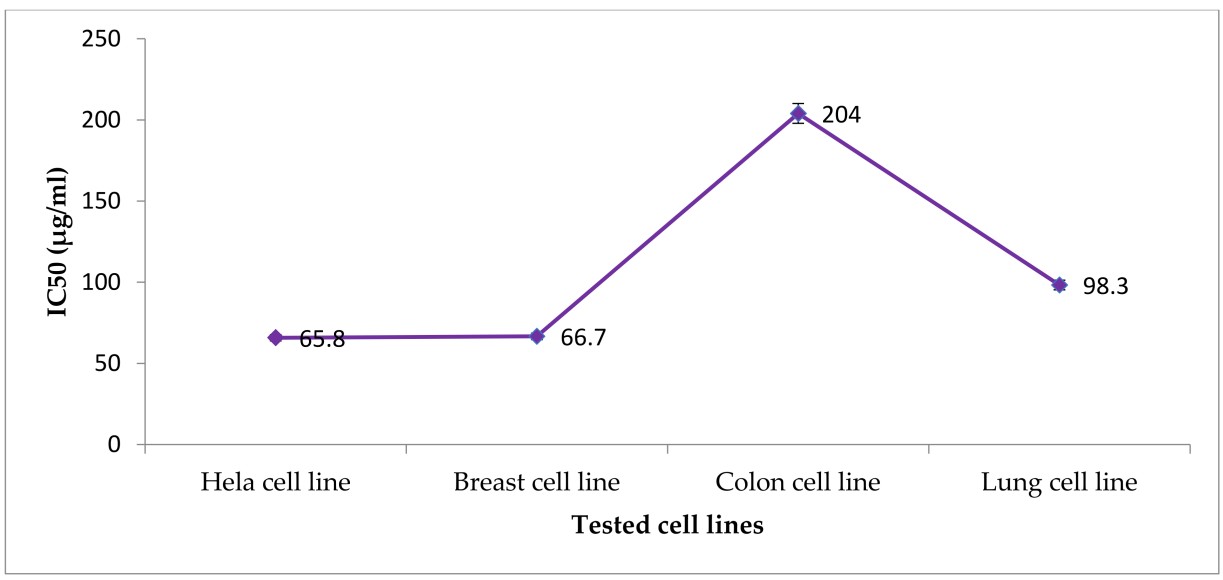

**Figure 3.** Antitumor (as $IC_{50}$) of tested extract (0.44 mg/L Zn of *Nostoc linckia)* against four cancer cell lines.

Figure 4 represents the percentage of anticancer activity of fractions (1, 6, 7 and 8) against cell line (Hela cell line). The $IC_{50}$ (μg/mL) of fractions (1, 6, 7 and 8) was 195.3, 258.9, 357.1 and 195.3 μg/mL, respectively.

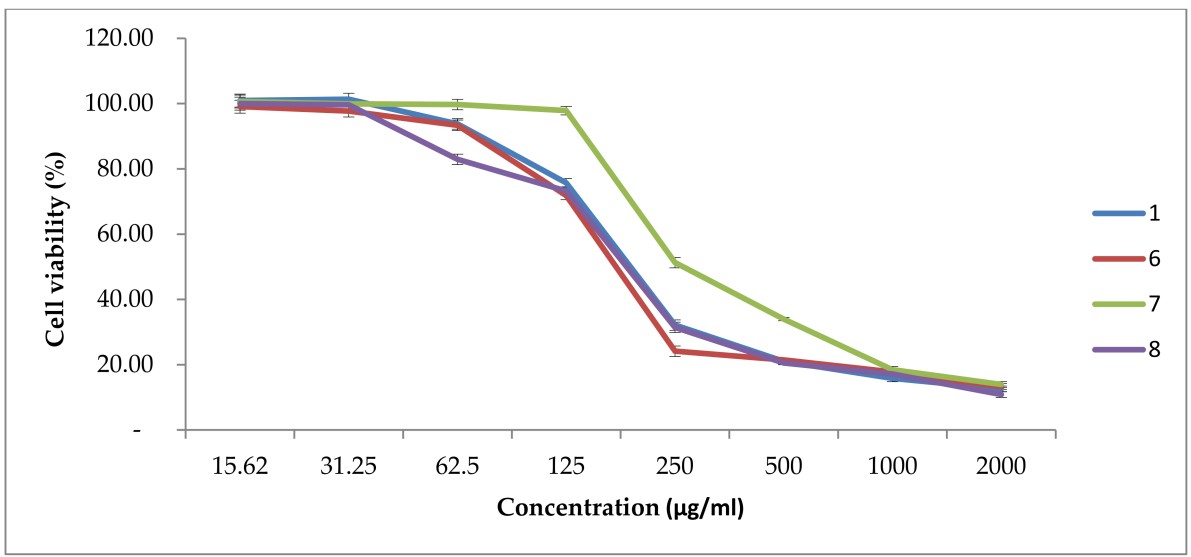

**Figure 4.** Antitumor activity (%) of the four fractions (1, 6, 7 and 8) from 0.44 mg/L (Zn) extract of *Nostoc linckia* against the promising cell line (Hela cell line).

## 4. Discussion

A greater understanding of cyanobacteria's heavy metal response mechanisms could lead to the development of these microorganisms for usage in the field of biologically active natural products. In our present work (Tables 3–5), the resistance of cyanobacteria to the high concentration of Cu can explain the increased growth rate at 0.079 mg/L Cu. For certain biomass enzymes, copper appears to control functional activity as a concentration factor [60]. In response to a wide range of divalent metal ion concentrations, some strains release mRNA for the stress protein GroEL and metallothionein's metal-binding protein. Following exposure to divalent cations at values ranging from 10 μM to 100 μM for Cu and Zn, a high transcription rate is established within 15 min. After one hour of exposure to every cation, transcript levels revert to normal. Within 15 min of these exposures, metallothionein operon is also induced. These resistance systems, we assume, operate collectively to protect the cell from injury [61]. Metallothioneins are cysteine-rich proteins that bind metal ions, reducing their cellular availability and ultimately detoxifying them [62]. Our findings were also consistent with those of Mamboya et al. [63], who found that copper is highly poisonous *to Padina boergesenii* at doses greater than 500 mg/L. A low concentration of copper, on the other hand, is harmful after 21 days of exposure. Because zinc works as a prosthetic moiety of several key enzymes that can readily impact metabolism, the observed differences in the growth rate of *Nostoc linckia* cultured under varying zinc concentrations (of BG-110 media) may be understood [64]. The noticed high growth rate of the cultures at 0.22 mg/L may be due to the tolerance of *Nostoc linckia* to this concentration. A fructose bisphosphate aldolase, DNA primase, carbonic anhydrase, alkaline phosphatase, and RNA polymerase are just a few of the zinc-dependent or binding proteins found in E.coli. TonB-dependent outer membrane proteins that implicated in the active transport of zinc or zinc chelates across membrane. Metals are also required as cofactors by around 40% of all enzymes, ranking from Mg (16%), Zn (9%), Fe (8%), Mn (6%), Ca (2%), Co and Cu (1%) [65]. On the other hand, zinc accumulation onto cells could explain the declined growth of cyanobacteria at higher zinc concentrations. In the end experiment time, this accumulation greatly affected/reduced the available zinc ion concentration and its toxicity to algal survival cells, which matched with the results [66].

Similarly, our findings (Table 5) revealed that a copper–zinc combination had a growth inhibiting impact when compared to control. The accumulation of metals (zinc or copper) onto cyanobacteria cells at higher concentrations significantly reduces the growth. At this concentration, zinc and copper ions led to toxicity to cyanobacterial cells during the final

experiment period [66]. Moreover, increased zinc and copper concentration resulted in a gradual decrease in water-soluble pigments contents. Heavy metals have also been shown to disrupt a range of photosynthetic activities [67]. Under standard conditions (molar concentrations; pH 7) and a half-cell potential (Eo) $\leq$ 816 mV, electron transfer from NADH (Eo =//$\geq$ 320 mV) releases around $\geq$ 238 kJ/mol enough to conserve three ATP/electron pairs transmitted. By electron transfer-dependent phosphorylation, no other respiratory electron acceptor allows for such significant energy conservation. We can conclude from our results that, all the metabolic processes are affected due to accumulation of Cu and Zn that reach the toxic level, the energy supply consequently decrease the growth rate decreased.

In cyanobacteria, phycobiliproteins are a category of colorful proteins that operate as light harvesters. Phycobiliproteins are water-soluble compounds made up of proteins and phycobilins, which are covalently bound via cysteine. Phycoerythrin, phycocyanin, phycoerythrocyanin, and allophycocyanin are the four primary types. Phycocyanin, phycoerythrin, and allo-phycocyanin are the most prevalent phycobiliproteins in cyanobacteria quantitatively [68]. In our present work (Table 6) the low concentration of Cu and Zn metals resulted in significant high production of phycobiliproteins. The membrane-bound P-type ATPases CtaA and PacS, as well as the metallochaperone Atx1, supply Cu to plastocyanin for photosynthesis and cytochrome c oxidase for respiration in Synechocystis PCC 6803 thylakoids [69]. Copper is required in cyanobacteria's thylakoids for type 1 copper site of plastocyanin, which is involved in photosynthetic electron transfer, as well as the CuA and CuB centers of cytochrome c oxidase, which is a terminal electron acceptor in aerobic respiration. The zinc exporter ZiaA (ZiaANmetal-binding)'s domain MBD has a nearly identical Cu(I) affinity. Thermodynamics of zinc removal from the cell may be influenced by the binding of potentially competing zinc to trafficking proteins, with ZiaAN having the highest Zn(II) affinity. Our results revealed that elevated concentrations of either Cu or Zn resulted in significant decrease in phycobiliproteins and the mechanism of metal toxicity, which depends mainly on the growth stage could explain that results. Our results agreed with many reports, they tested the effect of various concentrations of Pb(II) on *Spirulina platensis* conducted in Zarrouk liquid medium [70,71]. Phycobiliprotein contents were found to have decreased with the increased Pb(II) concentration at the end of 10-day culture period. A decrease in phycobiliprotein might be due to a change in the arrangement and structure of the photosystem II induced by led ions. The multiprotein complex (phycobilisome structure), which is localized in the thylakoid membrane, is composed of different phycobiliproteins. At a higher concentration of $Pb^{2+}$ the biosynthesis of membrane proteins dramatically affected, and this could be the possible reason for the decrease of phycobiliprotein contents. Similarly, heavy metals Cu and Zn may affect the phycobiliprotein in cyanobacterium culture. As copper is a heavy metal, the elevated concentration can act strongly on chromatin, photosynthetic apparatus, growth, pigments, and senescence processes.

Furthermore, Hemlata and Fatma discovered that in the presence of many metals ($Pb^{2+}$, $Cr^{6+}$, $Cu^{2+}$, $Zn^{2+}$, $Ni^{2+}$, and $Cd^{2+}$), the formation of phycobiliproteins by Anabaena NCCU-9 diminishes. The necessity for metabolic energy and amino acids for enzymatic defense explains this [72]. Fe and Cu can also generate Fenton and Fenton-like reactions, according to O'Halloran and Culotta [73], especially in oxygen-producing cyanobacteria. Metal chaperones are tiny proteins that bind transition metal cations in the cytoplasm or periplasm to reduce harmful reactions and interference with other metals, as well as to offer cations as cofactors to their specialized enzymes. Because copper is at the top of Irving-Williams family, plus a powerful thiol-binding metal and a toxic Fenton reagent, copper chaperones sequester copper in the cytoplasm of many species. This clearly explains our results by which the high decrease in phycobiliprotein content was recorded in elevated Cu treatments [74].

The level of secondary metabolites (Table 7) (phenolic, flavonoids and tannins) was elevated significantly at levels of 0.316 and 0.44 mg/L of Cu and Zn respectively. These results may be due to the effect of these metals concentrations on gene expression of

*Nostoc linckia* cells, which combined with enhancing biosynthesis of different secondary metabolites, mainly carotenoids, phenolic compounds and flavonoids. These findings corroborated those of Sakihama et al. [75], who discovered that these chemicals may serve as antioxidants as well as chelating agents [76]. These observations agreed with the previously published results [54,57,77]. Shafiq et al. [78] confirmed similar findings, stating that there was a strong link between antioxidant activity and total phenolic compound levels. Although secondary metabolites in cyanobacteria are widely studied, the effect of transitional metals, especially copper and zinc, still needs further research.

Antioxidant activity results of extracts in this study (Tables 8 and 9) were parallel with that of El-fayoumy et al. [5], who found that extract from Nostoc sp was the most effective extract, exhibiting a significant increase in antioxidant activity based on DPPH and ABTS assays (88.18 0.64% and 84.20 1.01%, respectively). Martel et al. [79] also tested DPPH radical scavenging capacity of different microalgae and cyanobacteria species in methanolic and aqueous extracts. They discovered that *Euglena cantabrica* has the strongest antiradical scavenging activity due to its high phenolic content. Moreover, Prabakaran et al. [80] evaluated the quantitative phytochemical profile of *C. vulgaris*. In total, seven phytochemicals were estimated quantitatively from methanolic extract, and phenols were found to be higher followed by the alkaloid, terpenoid and glycoside and the tannin were found to be lesser amount. Takyar et al. [81] examined the antioxidant activity of alcoholic extract of *C. vulgaris* as 42.96% at 200 ppm, while *Spirulina platensis* showed antioxidant activity (as 62.46% at 400 ppm) using DPPH method. These findings may be explained by heavy metals' mechanism and action as an inducer of oxidative stress in microalgal cells, which activated antioxidative enzymes as SOD, GST, CAT, and APX to reduce ROS [82]. According to [83,84], these biological reactions can be viewed as a tolerant mechanism. Antioxidant activity of active components in the effective extract of *Nostoc linckia* (0.44 mg/L Zn) might be attributed to several functional groups such as hydroxyl, sulfhydryl, and unsaturated bonds, which have a strong potential to scavenge free radicals and inhibit oxidation processes. The results of the extract's GC-MS analysis are associated with the data in Table 7, indicating that there is a strong relationship between antioxidant activity of effective extracts and flavonoids and phenolic compounds concentrations in Table 8. Siloxanes (cyclononasiloxane and cyclodecasiloxane), eneon compounds, esters, and phenyl esters are all found in the discovered substance. All those compounds were excepted an antioxidant activity.

The cyanobacteria grown under 0.44 mg/L Zn generated $21.05 \pm 0.92$ mg/g flavonoids and $89.29 \pm 1.46$ mg/g phenolic content, which recorded a higher combined antioxidant activity. Our findings matched those of Nedamani et al. [85], who looked at the antioxidant activity of individuals and combined rosemary and oak fruit extracts to detect possible interactions in their antioxidant activity. They found that there is a good relationship between total phenols and antioxidant activity. Rosemary extract exhibited a higher antioxidant activity (with higher entire phenol content) than oak extract. The combined extracts showed different behaviors with all three kinds of interactions (synergistic, antagonistic and additive effect). Singprecha et al. [86] investigated the effect of ginger extract and ascorbic acid on antioxidant activity, implying that an antagonistic interaction between ascorbic acid and ginger, or herbal extracts, should be considered to avoid any potential health consequences. El-Beltagi et al. [87], found that a mixture of Spirulina platensis and pomegranate juice ameliorate the rat hepatic harm caused by $CCl_4$ through its antioxidant activity.

In a pH range of 3–12, the extract's antioxidant activity was examined (Figure 1). Because of the presence of phenolic OH group, these compounds behaved as good antioxidants in solution. The primary mechanism of the chain-breaking action was linked to the hydrogen atom transfer (HAT) from the phenolic OH to peroxyl radicals since their antioxidant activity was inversely proportional to magnitude of their O−H bond dissociation enthalpies (BDE). The antioxidant activity of buffer solution was greatly influenced by its pH. All phenolic acids or esters were modest inhibitors of peroxidation in acid conditions (pH 3–5), but their antioxidant activity rose significantly with increasing pH.

The enhanced activity reported near the pKa value corresponding to the ionization of one of phenolic hydroxyl groups at pH 8-12 has been attributed to phenolate anion's high antioxidant activity [88].

Data showed that all fractions revealed lower activity than that exhibited by the crude extract (Table 12). Lower activities may be due to the antagonistic effect of crude extract components that cause an increment in ROS scavenging efficiency responsible for antioxidant activity. These results were in agreement with several reports [4,89,90]. The obtained data in Table 11 could be explained if we answer two questions. First, what is the relationship between the heavy metals' (Cu/Zn) stress and the antioxidants enzymes? The second question, how to join between lipid peroxidation and the levels of heavy metals and antioxidant enzymes? Many studies were joined between the metal ions and activity of antioxidant enzymes in cyanobacteria. The evolution of superoxide dismutase enzymes (SODs), which can remove superoxide free radicals, can be used to estimate when cyanobacteria first appeared. SODs containing copper and zinc cofactors were employed by cyanobacteria (CuZnSOD). Carotenoids, tocopherol, and antioxidant enzymes such as peroxidases, catalases, superoxide reductases (SORs), and superoxide dismutases (SODs) are used by cyanobacteria to eliminate ROS. SORs and SODs remove superoxide free radicals ($O_2^{\cdot-}$), whereas peroxidases and catalases speed up the removal of peroxides (such as $H_2O_2$ and R-O-O-H). Photosynthetic and respiratory electron transport chains 12, as well as extracellular processes on the cell surface, produce $O_2^{\cdot-}$ as a consequence [91–94]. They can also help with iron acquisition, cell signaling, and growth, but if $O_2^{\cdot-}$ is allowed to build up inside the cell, it reacts with solvent-exposed 4Fe-4S clusters in proteins, such as those involved in amino acid biosynthesis and photosynthesis, resulting in Fenton reaction reactants, which can cause extensive DNA damage. Organisms must keep control over their abundance to balance the good effects of $O_2^{\cdot-}$ with the damage caused by overexposure. SODs and SORs have been detected in all three domains of life—Eukarya, Archaea, and Bacteria—possibly because of this. For the first time, our phylogenetic analyses reveal CuZnSOD to be antioxidant enzyme that protects photosynthetic repair systems from oxidative damage. The reduced (thiol) sulfur atom (SH) of the cysteinyl residue of GSH catalyzes the nucleophilic attack on the electrophilic center of various hydrophobic compounds (R-X) by the GST superfamily (EC 2.5.1.18) [95]. Deglutathionylation has been discovered to play a role in photosynthetic organisms' redox regulation, protection, and recovery of oxidized enzymes in cyanobacterial cells. Lopez et al. [96] published findings that explained the effect of Cu on lipid peroxidation. Cu's propensity to cycle between oxidation states makes it a perfect cofactor in redox reactions, but it also allows Cu to catalyze the formation of reactive oxygen species via the metal catalyzed Haber-Weiss reaction, according to the researchers. •OH is produced as a result, causing severe lipid, protein, and DNA damage. The Irving-Williams series also predicts an alternative route of Cu toxicity, which is related to Cu's great capacity to bind organic compounds, which means Cu can displace other metals from their associated binding sites within metalloproteins [97,98]. Similarly, This interpretation applies to zinc because both Zn and Cu which are divalent cations have almost the same redox potential.

From the obtained data in Figures 2 and 3, the antitumor pattern of the *Nostoc linckia* crude extract of 0.44 mg/L Zn was similar in MCF-7, A549, and HeLa cell lines, while differrent on HCT 116 cells. The effect of the crude extract is concentration-dependent (31.25–500 g/mL), according to these findings. For these cells, the effect can be described as receptor-independent, as reported [99]. Furthermore, as previously noted in Table 10, the anticancer character of the crude extract may be related to its abundance of bioactive compounds. Anticancer compounds such as cycloheptasiloxane, hexadecanol, hexadecenoic acid, hexadecen, and octadecatrienoic acid methyl ester were abundant in cyanobacteria grown at 0.44 mg/L Zn [100–103]. The obtained results in Figure 4 revealed that the four promising fractions (1, 6, 7 and 8) from 0.44m g/L (Zn) of *Nostoc linckia* against the promising cell line (Hela cell line) have high antitumor effect ranged from 50 to 90% at 130 and 2000 µg/mL, respectively, with no significant difference between each of tested fraction

toward Hela cell line. The antitumor activity revealed in this investigation was similar to that identified by El-fayoumy et al. [5], who discovered that a crude extract of *Nostoc linckia* had anticancer activity against the HCT116 cell line with an $IC_{50}$ of 155 µg/mL. Moreover, Alghazeer et al. [104] who determined anticancer activity of crude extracts of some macroalgae species such as *Ulva lactuca*, *Cystoseira stricta* and *Sargassum vulgare* they reported that phenolic compounds (polyphenols and flavonoids), which are rich in crude promising extracts showed high antioxidant activity and antiradical characteristics. After exposing varied extract concentrations to $CaCO_2$ cells, anticancer activity was observed, and this activity was linked to polyphenol and flavonoid content. Obviously, both anticancer and antioxidant activity of crude extract of 0.44 mg/L Zn showed the highest activity compared to their promising fractions. Different behaviors of extract and its fractions can be attributed to the compounds' chemical properties, nature, and reactivity that may suffer polymerization, leading to structural changes responsible for the antioxidant activity.

### 5. Conclusions

Under copper and zinc stress, the growth rate of Cyanobacterium *Nostoc linckia* was examined. Copper and zinc concentrations higher than those found in the BG110 culture media were employed. We investigated potential antioxidant activity, antioxidant enzymes, anticancer and secondary metabolites, and performed lipid peroxidation. We can conclude from the obtained data that the antioxidant and anticancer results may encourage a comprehensive project to evaluate the activity *in-vivo*. Furthermore, our results support extensive research for use of the promising active extract *Nostoc linckia*, cultivated under 0.44 mg/L Zn as a source of different active constituents, especially for pharmaceutical remedies that raise the pharmaco-economic value of cyanobacteria in Egypt and any producing country. In addition, the tolerance of *Nostoc linckia* to grow under high concentrations of heavy metals may offer unique mechanisms to deposit the metal contaminants. This point needs further study to screen the crude extract with more specific spectroscopic methods to identify the absolute structure of the biologically active constituents.

**Author Contributions:** Conceptualization, K.M.A.R., H.S.E.-B, S.M.M.S., E.A.S. and E.S.A.B.; methodology, K.M.A.R., H.S.E.-B, E.A.E.-f., S.M.M.S., E.A.S. and E.S.A.B.; formal analysis, K.M.A.R., S.M.M.S., E.A.S., E.A.E.-f. and E.S.A.B.; investigation, K.M.A.R., H.S.E.-B, S.M.M.S. and E.A.S.; data curation, K.M.A.R., S.M.M.S., E.A.S., E.A.E.-f. and E.S.A.B.; writing—original draft preparation, K.M.A.R., H.S.E.-B, E.A.E.-f., S.M.M.S., E.A.S. and E.S.A.B.; writing—review and editing, K.M.A.R., H.S.E.-B, E.A.E.-f., S.M.M.S., E.A.S. and E.S.A.B.; supervision, K.M.A.R., H.S.E.-B, E.A.E.-f. and S.M.M.S. project administration, H.S.E.-B, S.M.M.S. and E.A.S. All authors have read and agreed to the published version of the manuscript.

**Funding:** This research was funded by the Deputyship for Research and Innovation, Ministry of Education in Saudi Arabia (project number IFT20185).

**Institutional Review Board Statement:** Not applicable.

**Informed Consent Statement:** Not applicable.

**Data Availability Statement:** Not applicable.

**Acknowledgments:** Authors extend their appreciation to Deputyship for Research & Innovation, Ministry of Education in Saudi Arabia for funding this research work through project number IFT20185.

**Conflicts of Interest:** The authors declare no conflict of interest.

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
