# Peer review of "Potential Antioxidant and Anticancer Activities of Secondary Metabolites of Nostoc linckia Cultivated under Zn and Cu Stress Conditions"

_processes, doi:10.3390/pr9111972_

Round 1

Reviewer 1 Report

The manuscript found that in heavy metals stress conditions the Nostoc linckia extracts have the ability of antioxidant and anticancer activities . This is an article with basic works, including design, preparation, characterization and in vitro experiments. The manuscript is general well written and easy to follow. But on the other hand, it still needs more experimental data to support the conclusion. I suggest resubmission after revision, my detailed suggestions are as follows:

  1. The title of the article mentioned various oxidative stress conditions but there are only Zn and Cu mentioned above, these seem like interstitial interactions, not a direct oxygen stress environment. I suggest doing something really oxygen-stressed, like treating with H2O2.
  2. The results of higher zinc and copper concentrations showed growth inhibition and 22 mg /L (Zn) and 0.079 mg/L (Cu) enhanced growth, more experiments are needed to confirm especially for the control group. For example, 0, 0.05, 0.1, 0.15, 0.2 mg /L (Zn) , 0.02, 0.04, 0.06, 0.08 mg/L (Cu) .
  3. “The extracts at 500µg/ml showed the lowest cell viability of the tested cell lines” needs more experiments to support, like 510, 520, 530, 540µg/ml.
  4. Why choose pH not other factors? Such as ions, temperature, growth hormone, light and the like.
  5. Table 6 Cu concentration of 0.79 mg /Land Zn concentration of 0 have exactly the same results?Whether there are miswrites?

Here are some minor details:

  1. In table 1 chemical formulas for some compounds do not have subscripts, and some figures do not retain consistent significant digits. Part of the temperature unit format is wrong, and some brackets have only left half without right half, check all.
  2. Figure 1 Lack of horizontal and vertical coordinates. Is it necessary to do a difference analysis? The title at the top of the picture should be removed.
  3. Figure 2, the cell viability is usually shown as a bar chart, Figure 3 the IC50 may be more intuitive to make a curve graph.
  4. Table 4, difference analysis of the seventh column was missing.

Author Response

Dear: Editor

First of all, Thanks for your great efforts

Kindly, please check the attached file (revised manuscript) and the following is a step by step response to reviewer comments

Thanks again

Best Regards,

Dr. Hossam El-Beltagi

Response to reviewer comments

No

Comments

Response

Reviewer No. 1

1

The title of the article mentioned various oxidative stress conditions but there are only Zn and Cu mentioned above, these seem like interstitial interactions, not a direct oxygen stress environment. I suggest doing something oxygen-stressed, like treating with H2O2.

As described in the discussion section, copper and Zinc in the normal range in algae culture are considered cofactors for some enzymes. However, at high metal concentration, the oxidation stress occurs by formation of ROS e.g., superoxide and hydroxyl radical via Fenton reaction (which correlated with the increase of some defense enzymes e.g., catalase and SOD enzymes) and increase the MDA (which mainly formed from oxidation of fatty acids). This type of stress on algal cells considers a biological oxidative stress causes the accumulation of some secondary metabolites as a kind of algal defense mechanisms such as phenolic compounds, flavonoids, and tannins because of stress on gene expression inside alga. Additionally, ROS end with formation of hydrogen peroxide in termination step in radical chain reaction. (Please review lines from 383 to 404 in discussion section)

These findings were in agreement with the many published results as following:

·         Cu stress decreased growth rate and pigment contents in microalgae (Sáeza et al., 2015; Machado and Soares, 2016) while increasing reactive oxygen species (ROS) generated through the interference of Cu ions in Fenton's reaction (Okamoto et al., 2001; Moenne et al., 2016; Machado and Soares, 2016; Olivares et al., 2016). Increased levels of ROS can rapidly attack nucleic acids, proteins, and lipids, leading to permanent metabolic dysfunction and cell death (Gill and Tuteja, 2010; Pandey et al., 2015). Cu stress-induced oxidative damages (e.g., augmentation of protein and lipid peroxidation) in Sce- nedesmus sp. and Spirulina platensis-S5, were previously recorded (Sabatini et al., 2009; Choudhary et al., 2007; Tripathi and Gaur, 2004; Tripathi et al., 2006; Jara et al., 2014; Sáeza et al., 2015; Olivares et al., 2016).

·         Significant increases in SOD activity and GSH content in Scenedesmus vacuolatus were recorded at high Cu levels (Sabatini et al., 2009) as a type of algal antioxidant defense enzymes mechanisms (Pinto et al., 2003; Mallick, 2004).

·         ‘Euglena gracilis treated with high conc. of Cu+2, increased significantly, the total phenolics content and total flavonoids concerning control (as reported by Gonzalez-Mendoza  Et al., 2013).

On the other hand, the treatments are Zn , Cu and a combination between two metallic ions at different levels hence the use of "various"

2

The results of higher zinc and copper concentrations showed growth inhibition and 22 mg /L (Zn) and 0.079 mg/L (Cu) enhanced growth, more experiments are needed to confirm especially for the control group. For example, 0, 0.05, 0.1, 0.15, 0.2 mg /L (Zn) , 0.02, 0.04, 0.06, 0.08 mg/L (Cu) .

We appreciate this suggestion. We would like to point out that the concentrations used in this manuscript was guided by the concentrations of exploratory experiments. We can work on the proposed concentrations in the future in a more focused study on this point.

3

“The extracts at 500µg/ml showed the lowest cell viability of the tested cell lines” needs more experiments to support, like 510, 520, 530, 540µg/ml

In the section the Double concentrations were used, where the next concentration was 1000 µg/ml, which gave an insignificant difference from the concentration of 500 µg/ml. as shown in Fig 2 the cell viability relatively stable in concentration range between 250-500 µg/ml.

4

Why choose pH not other factors? Such as ions, temperature, growth hormone, light and the like

We choose the effect of the change in pH as the factor that can change in the growing media or the extract due to the effect of microbial enzymes on the extract and thus test the stability and bioactivity of the extract, while the rest of the factors (which are important factors in affecting the activity of the extract) must be controlled externally and can be studied In an independent research, we study these factors and their interactions on the biological activity of the extract.

5

Table 6 Cu concentration of 0.79 mg /L and Zn concentration of 0 have exactly the same results? Whether there are miswrites?

We revised the row data, and it is not miswrites, the measured values were the same in both treatments.

Here are some minor details:

1

In table 1 chemical formulas for some compounds do not have subscripts, and some figures do not retain consistent significant digits. Part of the temperature unit format is wrong, and some brackets have only left half without right half, check all.

All mentioned corrections have been completed and please check the attached revised manuscript.

2

Figure 1 Lack of horizontal and vertical coordinates. Is it necessary to do a difference analysis? The title at the top of the picture should be removed.

done

3

Figure 2, the cell viability is usually shown as a bar chart, Figure 3 the IC50 may be more intuitive to make a curve graph.

done

4

Table 4, difference analysis of the seventh column was missing.

done

Reviewer 2 Report

The article text has been extensively highlighted in yellow, suggesting an already revised MS. 

My main comment has to do with the use of the term "total protein" in the 2.3.8. sub-section of the Materials and Methods section. Considering the statement  in line 112 "The crude extract was prepared by suspending the biomass in potassium phosphate buffer of pH 7 and ultrasonic cell disruption, followed by clarification of the extract by high-speed centrifugation (10,000 rpm)", the resulting aqueous supernatant contains mainly hydrophilic proteins. The hydrophobic ones (e.g. membraneous proteins) can not be extracted in aqueous phosphate-based solvents unless they are supplemented e.g. with detergents such as SDS. Therefore, the protein assay in the 2.3.8. sub-section does not quantify "total proteins", and the authors are advised to replace this term with "aqueous proteins"

Author Response

Dear: Editor

First of all, Thanks for your great efforts

Kindly, please check the attached file (revised manuscript) and the following is a step by step response to reviewer comments

Thanks again

Best Regards,

Dr. Hossam El-Beltagi

No

Comments

Response

Reviewer No. 2

1

My main comment has to do with the use of the term "total protein" in the 2.3.8. sub-section of the Materials and Methods section. Considering the statement in line 112 "The crude extract was prepared by suspending the biomass in potassium phosphate buffer of pH 7 and ultrasonic cell disruption, followed by clarification of the extract by high-speed centrifugation (10,000 rpm)", the resulting aqueous supernatant contains mainly hydrophilic proteins. The hydrophobic ones (e.g. membraneous proteins) can not be extracted in aqueous phosphate-based solvents unless they are supplemented e.g. with detergents such as SDS. Therefore, the protein assay in the 2.3.8. sub-section does not quantify "total proteins", and the authors are advised to replace this term with "aqueous proteins"

As we value the comment, the correction has been made please check the revised version of the manuscript.

Round 2

Reviewer 1 Report

I have recommended this paper to major revision. The reason is that the author's answers to relevant questions are not clear, no supplementary content for my questions, and there are still lots of details in the article, for example Figure 1 still lack of vertical and horizontal coordinates, Figure 2 the legend is incomplete, and compared to Table 7, the data in Table 6 have obvious problems, in abstract there is a bracket have only left half without right half, in line 233, Zn concentrations should be 0.44, 0.88 and 1.76 mg/L. From what has been discussed above the author still need to resubmit after substantial modifications.

Author Response

No

Comments

Response

Reviewer No. 1

1

The title of the article mentioned various oxidative stress conditions but there are only Zn and Cu mentioned above, these seem like interstitial interactions, not a direct oxygen stress environment. I suggest doing something oxygen-stressed, like treating with H2O2.

The title of the article has been modified to be fulfilled with reviewer comment please find that in the revised version of the manuscript.

Potential antioxidant and anticancer activities of secondary metabolites of Nostoc linckia cultivated under Zn and Cu stress conditions

2

The results of higher zinc and copper concentrations showed growth inhibition and 22 mg /L (Zn) and 0.079 mg/L (Cu) enhanced growth, more experiments are needed to confirm especially for the control group. For example, 0, 0.05, 0.1, 0.15, 0.2 mg /L (Zn) , 0.02, 0.04, 0.06, 0.08 mg/L (Cu).

We appreciate this suggestion. We would like to point out that the concentrations used in this manuscript were guided by the concentrations of exploratory experiments. Also our coauthors published other article in Biomass Conversion and Biorefinery journal

https://doi.org/10.1007/s13399-021-01509-7

3

“The extracts at 500µg/ml showed the lowest cell viability of the tested cell lines” needs more experiments to support, like 510, 520, 530, 540µg/ml

In the section the Double concentrations were used, where the next concentration was 1000 µg/ml, which gave an insignificant difference from the concentration of 500 µg/ml. as shown in Fig 2 the cell viability relatively stable in concentration range between 250-500 µg/ml. additionally, The results obtained from this part will be subject to further study so that close concentrations of the both original extract and fractions 1, 6, 7, 8, as well as the elucidation of these fractions with LC-MS to identify there active ingredient.

4

Why choose pH not other factors? Such as ions, temperature, growth hormone, light and the like

We chose the effect of the change in pH as the factor that can change in the growing media or the extract due to the effect of microbial enzymes on the extract and thus test the stability and bioactivity of the extract, while the rest of the factors (which are important factors in affecting the activity of the extract) must be controlled externally and can be studied In an independent research, we study these factors and their interactions on the biological activity of the extract.

5

Table 6 Cu concentration of 0.79 mg /L and Zn concentration of 0 have exactly the same results? Whether there are miswrites?

We revised the row data, and it is not miswrites, the measured values were the same in both treatments.

Here are some minor details:

1

In table 1 chemical formulas for some compounds do not have subscripts, and some figures do not retain consistent significant digits. Part of the temperature unit format is wrong, and some brackets have only left half without right half, check all.

All mentioned corrections have been completed and please check the attached revised manuscript.

2

Figure 1 Lack of horizontal and vertical coordinates. Is it necessary to do a difference analysis? The title at the top of the picture should be removed.

done

3

Figure 2, the cell viability is usually shown as a bar chart, Figure 3 the IC50 may be more intuitive to make a curve graph.

done

4

Table 4, difference analysis of the seventh column was missing.

done

Reviewer No. 1 , Round 2

1

I have recommended this paper to major revision. The reason is that the author's answers to relevant questions are not clear, no supplementary content for my questions, and there are still lots of details in the article, for example Figure 1 still lack of vertical and horizontal coordinates, Figure 2 the legend is incomplete, and compared to Table 7, the data in Table 6 have obvious problems, in abstract there is a bracket have only left half without right half, in line 233, Zn concentrations should be 0.44, 0.88 and 1.76 mg/L. From what has been discussed above the author still need to resubmit after substantial modifications.

1-      The previous answers were corrected and please check answers of comments 1,2 and 3 in the first part of this table.

2-      Figures 1 and 2 were modified according to the reviewer comment. Please check the revised manuscript.

3-      Tables 6 has been modified and please check the revised manuscript.

4-      The brackets in the abstract as well as in the whole paper was revised and corrected, please check the revised version of the manuscript

5-      Zn concentrations were corrected please check the revised manuscript.

This manuscript is a resubmission of an earlier submission. The following is a list of the peer review reports and author responses from that submission.

Round 1

Reviewer 1 Report

Authors showed that the Cyanobacterium Nostoc linckia treated with copper and zinc stress in culture was enhanced the antioxidant activity and antitumor activity involving induction of MAD and catalase. Overview the research results, some questions remained as follows.

・What is the nutrient stress and also how related to oxidative stress or cancer?

・Copper and Zinc are harmful heavy metals, thus they were reduced the growth involving protein content as indicated results. →Why MDA (lipid peroxidation) increased during increment of catalase and antioxidant activity with potential antioxidants (phenolics, flavonoids and tannins).

・Table 10 

→How much contain of phenolics, flavonoids and tannins in the active fractions (1, 6-8) and also how much contain the compounds in Table 8.

・Table 8

→How determined the relative percentage on GC/MS analysis. What use the internal standard?  In addition, authors should identify the compounds using authentic compounds.

・Fig. 3

→NO explanation of results of Fig. 3 in the text and spell out of cell viabikity (viability).

・Sentence of p12, 320-321 breaks off.

 As final comment, I will recommend that authors should more focus to find target molecule, then it will be more attractive research.

Author Response

Dear reviewer one

Thank you very much for your evaluation and comments on the manuscript titled "Chemical compositions, potential antioxidant, anticancer activities, and recommended industrial applications of semi-purified fractions of Nostoc linckia cultivated under various oxidative stress conditions". We have worked on your comments for improving the manuscript. Please find the attached file

Reviewer 2 Report

The research work entitled “Chemical compositions, potential antioxidant, anticancer activities and recommended industrial applications of semi-purified fractions of Nostoc linckia cultivated under various oxidative stress conditions” demonstrated that using micronutrients (copper and zinc) to stimulate Nostoc linckia to produce secondary metabolites with commercial value. The biological detection contained growth rate, content of secondary metabolites (i.e., phycobiliprotein, phenolic, flavonoids and tannins), antioxidant activity (including related enzyme activity), GC/MS analysis, protein content and anticancer activity. This work has forward-looking progress in exploiting the biological activity compounds of blue-green bacteria Nostoc linckia. Although I think that this work can make contribution for researchers to investigate the biological activity compounds of blue-green bacteria, the concepts and arguments proposed in the article is not presented well, Therefore, I suggested this work should undergo “major revision” before publishing in “Molecules”. Specific comments and general comments are given below:

Specific comments

  1. There are a lot of formatting errors in the article. Further English editing is necessary.
  2. The third paragraph of the introduction (Line 57-61), which author mentioned bioactive compounds of algae cells, is not very relevant to the previous paragraph. I suggest swapping this paragraph with the previous paragraph or deleting it directly.
  3. In the fourth paragraph of the introduction (Line 63-69), the author should also explain why copper and zinc are used as oxidative stimulants.
  4. Line 185-187, the meaning is unclear. If the authors attempt to explain the possible mechanism, the affected genes and the related metabolic pathways must be mentioned. Please rewrite this sentence.
  5. Line 324-325, …cyanobacteria species such as Ulva lactuca, Cystoseira stricta and Sargassum vulgare… Actually, these species belongs to marcoalgae (seaweeds).
  6. Section 2.6-2.8, The quality of writing here is unqualified. Rewriting is necessary. The author needs to add some contents, which include the comparison with other reports (g., any other strategies have used to improve the extraction process and increase the oxidizing ability of the extract).
  7. In material and method, all reagents and kits are not marked with company, city, and country.
  8. Among the Nostoc linckia extracts, which one is the most commercially potential bioactive compounds? The authors may give a conclusion.

General comments

  1. Line 45 and 48, Nostoc à Nostoc (Change the font to italic).
  2. Line 50, Nostoc sp. à Nostoc (Change the font to italic).
  3. Line 52, Cyanovirin à cyanovirin (Change to lowercase).
  4. Line 53, Influenza A à influenza A (Change to lowercase).
  5. Line 53, Nostoc sp. contains PUFAs that include… à Nostoc contains polyunsaturated fatty acids (PUFAs) that include. (The proper nouns that first appeared in the article need to be written in full).
  6. Line 57, Bioactive à bioactive (Change the font to italic).
  7. Line 61, … antitumor, antioxidant and antiviral activities…à … antitumor, antioxidant, and antiviral activities… (Put a comma in front of “and”).
  8. Line 68, …nutritional factors (eg. Nitrogen and carbon)…à nutritional factors (g., nitrogen and carbon)
  9. In Table 1 and Table 2, I suggest changing “Zero” in Table 1 and Table 2 to “Cu deprivation”; O.D àD. ; 1Xà 1 ´
  10. Line 104, Copper à copper (Change to lowercase).
  11. Line 107, 118, Zinc à zinc (Change to lowercase).
  12. Line 109, 137, 161, 178, 209, 292, 382, 606 p<0.05 à p < 0.05 (Change to italic; missing blank).
  13. Line 116, P. boergesenii à boergesenii (Change to italic)
  14. Line 117, Nostoc linckia à Nostoc linckia (Change to italic)
  15. Line 130, cyanobacterialcells à cyanobacterial cells (Missing blank).
  16. Line 217, IC50 à IC50 (subscript)
  17. 3, Wrong text on y-axis, viabikity à viability (Misspelling)
  18. Line 288-289, These results were in agreement with the results obtained by El-fayoumy et al.; Afify et al., Younes et al. [4, 50, 51] à These results were in agreement with many reports [4, 50, 51].

Author Response

Dear reviewer 2

Thank you very much for your evaluation and comments on the manuscript titled " Chemical compositions, potential antioxidant, anticancer activities, and recommended industrial applications of semi-purified fractions of Nostoc linckia cultivated under various oxidative stress conditions". We have worked on your comments for improving the manuscript. Please find the attached file

Reviewer 3 Report

The authors inspect the effect of metal stressors like Cu and Zn on the growth and metabolite production of a cyanobacterium from the genus Nostoc. The article evidences a large number of data obtained from this cyanobacterium placed in culture. However, this work suffers from important shortcomings that prevent its publication as such. The text is extremely difficult to read due to a low level of written English. The authors should take more care to the style and presentation as there are for example too many tables with a large amount of data. Importantly the chemical content should also be analysed by LC-MS as GC-MS only reveals volatile compounds and the bioactive components might not be volatile.

Author Response

Dear reviewer 3

Thank you very much for your evaluation and comments on the manuscript titled " Chemical compositions, potential antioxidant, anticancer activities, and recommended industrial applications of semi-purified fractions of Nostoc linckia cultivated under various oxidative stress conditions". We have worked on your comments for improving the manuscript. Please find the attached file

Round 2

Reviewer 1 Report

Although authors sincerely replied, but the answers are not yet complete or insufficient to my questions. The manuscript should not be separated (No7). As I suggested in previous comments, I will recommend that authors should more focus on insight the molecule, then it will be suitable version for the scope of Molecule.

Reviewer 2 Report

Reviewer’s response

Title: Chemical compositions, potential antioxidant, anticancer activities and recommended industrial applications of semi-purified fractions of Nostoc linckia cultivated under various oxidative stress conditions

Journal: Molecules

In my personal opinion, the author's response did not make this article meet the requirements of the journal, and there are still many formatting errors in the article. I suggested this work still need a lot of improvements before it could be accepted for publication in “Molecules”.

Comments for authors’ responses

  1. There are still many formatting errors. Further English editing is necessary.
    • Line 49, field for fistula treatment and cancer therapy [8] and anti-inflammatory…  field for fistula treatment, cancer therapy [8], anti-inflammatory… (remove unnecessary conjunctions).
    • Line 55, ?-linolenic acid [ 26].  ?-linolenic acid [26]. (remove blank)
    • Line 56, tocopherols,and… tocopherols, and… (missing blank)
    • Line 69-71, font format error
    • Line 68, (g., Nitrogen, and carbon)  (g., nitrogen and carbon)
    • Line 306, …as 195.3, 258.9, 357.1 and 195.3µg/ml respectively  as 195.3, 258.9, 357.1, and 195.3 µg/ml, respectively (missing comma and blank)
    • Line 323, 2000 ug/ml  2000 μg/ml
    • Line 502, 354nm  354 nm (missing blank)
  2.  
  3. Line 69-71, I think the author's response still needs to be strengthened. I suggest that the author explain in advance that various heavy metals can cause the physiological effects of photosynthetic organisms, and then elaborate on the reasons for choosing Cu and Zn.
  4. Okay
  5. Okay
  6. Section 2.6-2.8, The quality of writing here is still unqualified. Although the author has strengthened the exposition of experimental conditions and results, there is still nothing about comparison with other reports related to improve the extraction process and increase the oxidizing ability of the extract. Please note there are many formatting errors and grammatical error in these 3 sections. A comprehensively rewriting is necessary.
  7. Line 499, Cyanobacteria homogenate + phosphate buffer (pH 7.5)…; Line 500, Supernatant + chromogen…The meaning is unclear. Please rewriting the sentences.

Reviewer 3 Report

No clear improvement